# The Selective $G$-Bispectrum and its Inversion: Applications to $G$-Invariant Networks

**Simon Mataigne**[*]
ICTEAM, UCLouvain
Louvain-la-Neuve, Belgium
`simon.mataigne@uclouvain.be`

**Johan Mathe**
Atmo
San Francisco, CA
`johan@atmo.ai`

**Sophia Sanborn**
Science
San Francisco, CA
`sophiasanborn@gmail.com`

**Christopher Hillar**
Algebraic
San Francisco, CA
`hillarmath@gmail.com`

**Nina Miolane**[†]
UC Santa Barbara
Santa Barbara, CA
`ninamiolane@ucsb.edu`

## Abstract

An important problem in signal processing and deep learning is to achieve *invariance* to nuisance factors not relevant for the task. Since many of these factors are describable as the action of a group $G$ (e.g., rotations, translations, scalings), we want methods to be $G$-invariant. The $G$-Bispectrum extracts every characteristic of a given signal up to group action: for example, the shape of an object in an image, but not its orientation. Consequently, the $G$-Bispectrum has been incorporated into deep neural network architectures as a computational primitive for $G$-invariance—akin to a pooling mechanism, but with greater selectivity and robustness. However, the computational cost of the $G$-Bispectrum ($\mathcal{O}(|G|^2)$, with $|G|$ the size of the group) has limited its widespread adoption. Here, we show that the $G$-Bispectrum computation contains redundancies that can be reduced into a *selective $G$-Bispectrum* with $\mathcal{O}(|G|)$ complexity. We prove desirable mathematical properties of the selective $G$-Bispectrum and demonstrate how its integration in neural networks enhances accuracy and robustness compared to traditional approaches, while enjoying considerable speeds-up compared to the full $G$-Bispectrum.

## 1   Introduction

The visual world is rich with symmetries. For example, the identity of an object is invariant to its position in the visual field; vision has *translational symmetry*. Group theory is the mathematics used to describe transformations, their actions on objects, and the object's symmetry. As such, group theory has penetrated the fields of signal processing and deep learning alike. For example, the Fourier transform, pillar of signal processing, has been adapted to the $G$-Fourier transform, with its spectrum decomposing a signal defined over a group into several frequencies. More recently, researchers have become interested in the properties of higher-order spectra such as the *Bispectrum*, and its generalization to signals over groups via the $G$-Bispectrum.

$G$-**Bispectrum**   The $G$-Bispectrum is the Fourier transform of the $G$-Triple Correlation ($G$-TC). Historically, higher-order spectra found initial applications in the context of classical signal processing as generalizations of the two-point autocorrelation [31, 2, 25]. The work of Kakarala [15] illuminated

---

[*]Simon Mataigne is a Research Fellow of the Fonds de la Recherche Scientifique - FNRS.
[†]Nina Miolane acknowledges funding from the NSF grant 2313150.

the relevance of the $G$-Bispectrum for invariant theory, as it is the lowest-degree spectral invariant that is complete. Since then, it has appeared in diverse settings such as vision science [34], machine learning [19, 20], and 3D modeling [18].

**Limitations of the $G$-Bispectrum for Deep Learning**   The computational complexity of the $G$-Bispectrum has severely limited the reach of its applications. The most salient example of this limitation is in machine learning and deep learning. Convolutional Neural Network (CNN) [22, 24] reflect and exploit the translational symmetry of the visual world. Group-Equivariant CNNs ($G$-CNNs) [6, 21] do just this, with more general group-equivariant convolutions to exploit symmetries like rotational symmetries. In both cases, one typically wants to preserve transformations throughout the layers of a network (i.e., to be group-*equivariant*), and remove them only at the end when "canonicalizing" an image for classification (i.e., to be group-*invariant*). While the theory of equivariant layers has been thoroughly developed [7, 33], less attention has been paid to the theory of invariant layers [12]. This is where the $G$-Bispectrum enters the picture, and where its computational cost has strongly limited its integration into deep learning.

Commonly, invariance in $G$-CNNs is achieved by simply taking an average or maximum over the transformation group (Average or Max $G$-Pooling, respectively). However, as noted by Sanborn & Miolane [28], this is a highly lossy operation removing information about the structure of the signal. While the `max` operation is indeed invariant (the max of an image is the same as the `max` of an image rotated by 90 degrees), it is *excessively* invariant: one could permute all of the pixels in the image and without changing the maximum, but with none of the same structure (see Figure 8). To address this, Sanborn & Miolane [28] used the $G$-TC as a $G$-invariant layer that is *complete*—that is, it removes group transformations with no loss of signal structure. This approach achieves demonstrable gains in accuracy and robustness [28], but it is computationally expensive.

Indeed, the space complexity of the $G$-TC, i.e, its number of coefficients, scales as $\mathcal{O}(|G|^2)$, where $|G|$ is the size of the group. As each coefficient demands for $\mathcal{O}(|G|)$ operations, the computational cost or the time complexity of the $G$-TC is $\mathcal{O}(|G|^3)$. An alternative would be to use the $G$-Bispectrum as the pooling layer. However, both its space and time complexities are $\mathcal{O}(|G|^2)$. By comparison, the Max $G$-pooling layer features a $\mathcal{O}(|G|)$ computational cost and returns a scalar output. This raises the question of whether one can achieve complete invariance and adversarial robustness without sacrificing too much in terms of computational efficiency.

**Contributions.**   In this work, we prove for the first time that we can significantly reduce the computational complexity of the $G$-Bispectrum. This result has important implications for signal processing and deep learning on groups, for which the $G$-Bispectrum is a foundational computational primitive. Our contributions are:

- We provide a general algorithm that reduces the computational complexity of the $G$-Bispectrum from $\mathcal{O}(|G|^2)$ to $\mathcal{O}(|G|)$ in space complexity and from $\mathcal{O}(|G|^2)$ to $\mathcal{O}(|G|\log|G|)$ in time complexity if an FFT is available on $G$. We term it the *selective $G$-Bispectrum*. The algorithm can be applied to any finite group.

- We prove that the selective $G$-Bispectrum is complete for the most important finite groups used in practice, i.e., all discrete commutative groups, the dihedral groups of any order, the octahedral and full octahedral group. This significantly extends the work of [10, 15, 27], who first showed this for *some* finite, commutative groups, where it was demonstrated that the $G$-Bispectrum can be computed with only $|G|$ space complexity.

- We use the selective $G$-Bispectrum to propose a new $G$-*invariant* layer that strikes a balance between robustness and efficiency. In particular, it is more expensive than the Max $G$-pooling, but cheaper than the $G$-TC pooling.It is also cheaper than the full $G$-bispectral pooling of $\mathcal{O}(|G|^2)$ time and $\mathcal{O}(|G|^2)$ space complexity. The selective $G$-Bispectrum is more robust than the max $G$-pooling, and almost as robust as the $G$-TC.

- We run extensive experiments on the MNIST [23] and EMNIST [5] datasets to evaluate how each invariant layer (Max $G$-pooling, $G$-TC, selective $G$-Bispectrum) impacts accuracy and speed on classification tasks. We achieve the expected results: Our layer is faster than the $G$-TC and full $G$-Bispectrum and more accurate than Max $G$-pooling.

- We present several findings important to the design of invariant layers to guide further advances in the field of geometric deep learning. In particular, we show that the accuracy and speed advantages

of the selective $G$-Bispectrum is most striking for $G$-CNNs with low number of convolutional filters. Conversely, increasing the number of filters in the $G$-Convolutions allows the Max $G$-Pooling to catch up on the accuracy. This demonstrates that the $G$-bispectral pooling will be particularly interesting for neural networks operating under a smaller parameter budget.

We hope that the proposed reduction of the $G$-Bispectrum complexity will further open areas of research in signal processing on groups, that were previously prohibited due to the high complexity of the operation.

## 2 Background: $G$-Triple Correlation and $G$-Bispectrum

The proposed selective $G$-Bispectrum operation is closely related to two other foundational operations on signals defined on groups: the $G$-Triple Correlation and the full $G$-Bispectrum, which we introduce here. The background on group theory, including the definitions of groups, group actions, equivariance and invariance, is presented in Appendix A.

**The $G$-Triple Correlation**    Given a real signal defined on a finite group $\Theta : G \mapsto \mathbb{R}$, the $G$-Triple Correlation ($G$-TC) [15] is the lowest order polynomial that is complete, i.e., that conserves all of the information of the signal $\Theta$, up to group action by $G$.

**Definition 2.1.** The $G$-*Triple Correlation* of a real signal $\Theta : G \mapsto \mathbb{R}$ is given by

$$T(\Theta)_{g_1,g_2} := \sum_{g \in G} \Theta(g)\Theta(g \cdot g_1)\Theta(g \cdot g_2) \text{ for all } g_1, g_2 \in G. \tag{1}$$

The original triple-correlation was introduced for the classical framework of translations of a one-dimensional signal, i.e., where $X = \mathbb{Z}$ and $(G, \cdot) = (\mathbb{Z}, +)$. The $G$-triple correlation from Definition 2.1 extends the original definition to any finite group $(G, \cdot)$. In our setting, the signal $\Theta : G \mapsto \mathbb{R}$ will be obtained after the $G$-convolution of a function $f : X \mapsto \mathbb{R}^c$, representing a continuous image with $c$ channels, with a filter $\phi : X \mapsto \mathbb{R}^c$, and the $G$-TC will be applied channel-by-channel. Importantly, the $G$-TC layer has computational complexity $\mathcal{O}(|G|^3)$ and outputs $\mathcal{O}(|G|^2)$ coefficients.

**The $G$-Bispectrum**    The $G$-TC operation has a Fourier equivalent: the $G$-Bispectrum. Indeed, the definition of the Discrete Fourier Transform (DFT) can be extended to any finite group (see, e.g., [9]), as recalled below.

**Definition 2.2.** Given a set of unitary representatives (Def. A.2) of the equivalence classes of irreps $\rho_i : G \mapsto \mathrm{GL}(V_i)$ (Def. A.6), the $G$-*Fourier Transform* on a finite group $G$ of a signal $\Theta : G \mapsto \mathbb{R}$ is defined as

$$\mathcal{F}(\Theta)_{\rho_i} := \sum_{g \in G} \Theta(g)\rho_i(g)^\dagger, \tag{2}$$

where $\rho_i(g)^\dagger$ refers to the conjugate transpose of the matrix $\rho_i(g)$ (or simply transpose if $\rho_i$ is real-valued).

The $G$-Bispectrum $\beta(\Theta)$ is defined as $\mathcal{F}(T(\Theta))$, with $\mathcal{F}$ evaluated over the group $G \times G$. Kakarala [15] proposed a closed-form expression for the $G$-Bispectrum $\beta(\Theta)$ directly in terms of $\mathcal{F}(\Theta)$. We recall it in Theorem 2.3.

**Theorem 2.3.** *[17] The $G$-Bispectrum of a signal $\Theta : G \mapsto \mathbb{R}$, $\beta(\Theta)$, is given by:*

$$\beta(\Theta)_{\rho_1,\rho_2} = [\mathcal{F}(\Theta)_{\rho_1} \otimes \mathcal{F}(\Theta)_{\rho_2}] \, C_{\rho_1,\rho_2} \left[ \bigoplus_{\rho \in \rho_1 \otimes \rho_2} \mathcal{F}(\Theta)_\rho^\dagger \right] C_{\rho_1,\rho_2}^\dagger,$$

*where $C_{\rho_1,\rho_2}$ is a unitary matrix called the Clebsch-Gordan matrix, whose definition is recalled in Appendix A. For each pair $\rho_1, \rho_2$, the matrix $\beta(\Theta)_{\rho_1,\rho_2}$ is called a $G$-bispectral coefficient.*

For commutative groups, Theorem 2.3 simplifies to a more compact expression, recalled in Theorem A.12. However, both the space and time complexity of the $G$-Bispectrum remain $O(|G|^2)$.

To the authors' best knowledge, there is no generic analytical formula for computing $C_{\rho_1,\rho_2}$ for an arbitrary group $G$. However, there exist formulas for specific classes of groups (see Appendix E.2). Additionally, there exist packages for computing these for many groups using packages such as `escnn` [3].

Figure 1: Illustration of the different proposed $G$-CNN modules [6, 28]. The input $f$ is first processed through the $G$-convolutional layer composed of $K$ filters $\{\phi_k\}_{k=1}^{K}$. Then, an invariant layer is chosen (Max $G$-pooling, $G$-TC, or the selective/full $G$-Bispectrum layer). Finally, the "pooled" output is fed to a neural network designed for the machine learning task at hand.

**Complete $G$-Invariants**  The $G$-TC and the $G$-Bispectrum are desirable computational primitives for signal processing and deep learning because they are *complete $G$-invariants* (for generic data $\Theta, \widetilde{\Theta}$). Indeed, this completeness property make them very interesting for building invariant layers in $G$-CNNs, as they are selectively invariant. We define *complete $G$-invariance* next.

**Theorem 2.4.** *[16, Thm.3.2] The $G$-TC and the $G$-Bispectrum are complete $G$-invariants, i.e., for $\Theta, \widetilde{\Theta} : G \mapsto \mathbb{R}$ with $\mathcal{F}(\Theta)_\rho$ nonsingular for all irreps $\rho$, $T(\Theta) = T(\widetilde{\Theta})$, respectively $\beta(\Theta) = \beta(\widetilde{\Theta})$, if and only if there exists $h \in G$ such that $\Theta(g) = \alpha(h, \widetilde{\Theta}(g))$ for all $g \in G$.*

**Application: $G$-invariant layers**  The $G$-CNN architecture, first proposed in [6], is illustrated in Figure 1. The input signal $f : X \mapsto \mathbb{R}$, typically an image, is processed through a $G$-Convolution layer using filters $\{\phi_k\}_{k=1}^{K}$. The output is feature maps $\{\Theta_k\}_{k=1}^{K}$ that form a set of $K$ real-valued signals with domain $G$. This $G$-Convolution layer is traditionally followed by a $G$-invariant layer. The most common is the Max $G$-Pooling layer. More recent works have proposed two alternatives based on the $G$-TC and the full $G$-Bispectrum: the $G$-TC Pooling [28] and the (full) $G$-Bispectrum [29] respectively, where the latter requires the computations of the Fourier transforms of the feature maps, preferably computed using a Fast Fourier Transform (FFT) algorithm on $G$ [9]. When testing the impact of the choice of $G$-invariant layer, the output of the invariant layer is typically fed to a Secondary Neural Network (NN) to perform the desired task, e.g., image classification. The Secondary NN often takes the form of a Multi-Layer Perceptron (MLP).

Experimental results have demonstrated the superior accuracy and adversarial robustness of the $G$-CNN equipped with a $G$-TC and $G$-Bispectrum invariant layer [29, 28]. However, both methods inherit the high space and time complexity of their respective operations. This raises the question of whether we can reduce this computational complexity.

## 3    Method: The Selective $G$-Bispectrum and its Inversion

**The Selective $G$-Bispectrum**  We introduce a novel tool for signal processing on groups: the selective $G$-Bispectrum. A selective $G$-Bispectrum $\beta_{sel}$ is any $\mathcal{O}(|G|)$ subset of all coefficients of the $G$-Bispectrum $\beta$ (Definition 2.3), only conserving well-chosen pairs of irreps $(\rho_1, \rho_2)$. Which pairs of irreps to select depends on the group of interest. This is possible due to redundancies and symmetries in the full object. Below, we provide an algorithmic procedure to compute the selective $G$-Bispectrum for any finite group $G$ that features at most $|\text{Irreps}|(\leq |G|)$ coefficients. The procedure is summarized in Algorithm 1. We have the following proposition.

**Proposition 3.1.** *The selective $G$-Bispectrum $\beta_{sel}$ from Algorithm 1 has at most $|G|$ coefficients.*

*Proof.* By construction of Algorithm 1, $|L_\rho| \leq |\text{Irreps}|$. Since $|\text{Irreps}|$ is at most the number of conjugacy classes of $G$ (see, e.g., Steinberg [30, Corollary 4.3.10]), we have $|\text{Irreps}| \leq |G|$.    □

In Algorithm 1, we note that the choice of $\rho_1$ is important, since some $\rho_1$ will not allow the user to recover all of the irreps and therefore not ensure the completeness of the selective. We illustrate the computation of the selective $G$-Bispectrum in Figure 2, where we choose $\tilde{\rho}_1 = \rho_6$.

**Algorithm 1** Selective $G$-Bispectrum on any finite group $G$

1: **Input**: Signal $\Theta : G \mapsto \mathbb{R}$ with $n = |G|$. Empty list of coefficients $L_\beta$. Empty list of irreps $L_\rho$. Kronecker Table of $G$.
2: Add $\beta(\Theta)_{\rho_0,\rho_0}$ to $L_\beta$ where $\rho_0$ is the trivial irreps ($\rho_0(g) = 1$ for all $g \in G$).
3: Add $\rho_0$ to $L_\rho$.
4: Choose $\tilde{\rho}_1$ such that $\tilde{\rho}_1 \otimes \tilde{\rho}_1 = C_{\tilde{\rho}_1,\tilde{\rho}_1} \left( \bigoplus_{\rho \in \mathcal{R}} \rho \right) C_{\tilde{\rho}_1,\tilde{\rho}_1}^\dagger$ generates at least one irreps $\rho$ not yet in $L_\rho$.
5: Add $\beta(\Theta)_{\rho_0,\tilde{\rho}_1}$ and $\beta(\Theta)_{\tilde{\rho}_1,\tilde{\rho}_1}$ to $L_\beta$, add $\tilde{\rho}_1$ to $L_\rho$.
6: Add every $\rho$ that appears in $\tilde{\rho}_1 \otimes \tilde{\rho}_1$ to $L_\rho$.
7: **while** $L_\rho$ keeps changing: **do**
8:     Find $\rho'$, $\rho''$ in $L_\rho$ such that $\rho' \otimes \rho''$ generates at least one irreps not already in $L_\rho$.
9:     Add $\beta(\Theta)_{\rho',\rho''}$ to $L_\beta$.
10:     Add $\rho$ that appears in $\rho' \otimes \rho''$ to $L_\rho$.
11: **end while**
12: **Return** $\beta_{sel}(\Theta) := L_\beta$: the selected $G$-bispectral coefficients.

Computation of the Selective Bispectrum for the full octahedral group
Super-imposed on top of its Kronecker Table

| $\otimes$ | $\rho_0$ | $\rho_1$ | $\rho_2$ | $\rho_3$ | $\rho_4$ | $\rho_5$ | $\rho_6$ | $\rho_7$ | $\rho_8$ | $\rho_9$ |
|---|---|---|---|---|---|---|---|---|---|---|
| $\rho_0$ | 1000000000 | 0100000000 | 0010000000 | 0001000000 | 0000100000 | 0000010000 | 0000001000 | 0000000100 | 0000000010 | 0000000001 |
| $\rho_1$ | 0100000000 | 1111000000 | 0111000000 | 0110000000 | 0010000000 | 0000001000 | 0000011110 | 0000001100 | 0000001100 | 0000000100 |
| $\rho_2$ | 0010000000 | 0111000000 | 1111000000 | 0110000000 | 0100000000 | 0000000100 | 0000001111 | 0000011110 | 0000001100 | 0000001000 |
| $\rho_3$ | 0001000000 | 0110000000 | 0110000000 | 1001100000 | 0001000000 | 0000000010 | 0000001100 | 0000001100 | 0000010011 | 0000000010 |
| $\rho_4$ | 0000100000 | 0010000000 | 0100000000 | 0001000000 | 1000000000 | 0000000001 | 0000000100 | 0000001000 | 0000000010 | 0000010000 |
| $\rho_5$ | 0000010000 | 0000001000 | 0000000100 | 0000000010 | 0000000001 | 1000000000 | 0100000000 | 0010000000 | 0001000000 | 0000100000 |
| $\rho_6$ | 0000001000 | 0000011110 | 0000001111 | 0000001100 | 0000000100 | 0100000000 | 1110000000 | 0111100000 | 0110000000 | 0010000000 |
| $\rho_7$ | 0000000100 | 0000001111 | 0000011110 | 0000001100 | 0000001000 | 0010000000 | 0111100000 | 1111000000 | 0110000000 | 0100000000 |
| $\rho_8$ | 0000000010 | 0000001100 | 0000001100 | 0000010011 | 0000000010 | 0001000000 | 0110000000 | 0110000000 | 1001100000 | 0001000000 |
| $\rho_9$ | 0000000001 | 0000000100 | 0000001000 | 0000000010 | 0000010000 | 0000100000 | 0010000000 | 0100000000 | 0001000000 | 1000000000 |

Selective Bispectrum  $\{\beta_{\rho_0,\rho_0}, \beta_{\rho_6,\rho_0}, \beta_{\rho_6,\rho_6}, \beta_{\rho_1,\rho_2}, \beta_{\rho_1,\rho_6}, \beta_{\rho_1,\rho_7}\}$

Figure 2: Computation of the selective $G$-Bispectrum for the Full Octahedral Group. The gradient of color represents the order in which the $G$-bispectral coefficients are computed. The Kronecker Table represents which irreps emerge from the decomposition into irreps of the tensor product $\rho_i \otimes \rho_j$. We observe that the selective $G$-Bispectrum has only 6 coefficients, compared to 100 coefficients for the full $G$-Bispectrum.

**Inverting the Selective $G$-Bispectrum for completeness**  The *inversion* of the selective $G$-Bispectrum $\beta_{sel}(\Theta)$ is reconstructing a signal $\mathcal{F}(\widetilde{\Theta})$ from the $G$-Bispectrum coefficients in the list $L_\beta$ such that $\widetilde{\Theta} = \alpha(g, \Theta)$ for some $g \in G$ (The $G$-Bispectrum is $G$-invariant, hence, $\Theta$ can only be recovered at best up to group action). Once $\mathcal{F}(\widetilde{\Theta})$ is known, $\widetilde{\Theta}$ can be obtained using the Inverse Fourier Transform. If the selective $G$-Bispectrum can be inverted, then, by definition, it is complete in the sense of Theorem 2.4.

## 4    Theory: Completeness of the Selective $G$-Bispectrum

Our main theoretical claim is that the selective $G$-Bispectrum can be inverted and is a complete $G$-invariant that drastically reduces the complexity of the $G$-Bispectrum. We prove this claim for many finite groups $G$ of interest in signal processing and deep learning in a sequence of theorems presented in this section.

**Known Theorems**  Previous authors had looked into the $G$-Bispectrum inversion problem. It is well known that $|G| = n$ coefficients are enough for the cyclic group $(C_n, \cdot) = (\mathbb{Z}/\mathbb{Z}_n, + \mod n)$.

**Theorem 4.1.** *[17] For cyclic groups $C_n$, $n \in \mathbb{N}_0$, the $C_n$-Bispectrum can be inverted using $|G| = n$ coefficients if $\mathcal{F}(\Theta)_\rho \neq 0$ for all irreps $\rho$ of $C_n$.*

Similarly for a product of two such groups, we have the following theorem.

**Theorem 4.2.** *[10] For a product of cyclic groups $C_n \times C_m$, $n, m \in \mathbb{N}_0$, the $G$-Bispectrum can be inverted using $|G| = nm$ coefficients if $\mathcal{F}(\Theta)_\rho \neq 0$ for all $\rho \in C_n \times C_m$.*

Full and Selective Bispectra for the group $D_4$
Super-imposed on top of $D_4$'s Kronecker Table

| $\otimes$ | $\rho_0$ | $\rho_1$ | $\rho_2$ | $\rho_3$ | $\rho_4$ |
|---|---|---|---|---|---|
| $\rho_0$ | 10000 | 01000 | 00100 | 00010 | 00001 |
| $\rho_1$ | 01000 | 10000 | 00010 | 00100 | 00001 |
| $\rho_2$ | 00100 | 00010 | 10000 | 01000 | 00001 |
| $\rho_3$ | 00010 | 00100 | 01000 | 10000 | 00001 |
| $\rho_4$ | 00001 | 00001 | 00001 | 00001 | 11110 |

Full and Selective Bispectra for the group $O_h$
Super-imposed on top of $O_h$'s Kronecker Table

| $\otimes$ | $\rho_0$ | $\rho_1$ | $\rho_2$ | $\rho_3$ | $\rho_4$ |
|---|---|---|---|---|---|
| $\rho_0$ | 10000 | 01000 | 00100 | 00010 | 00001 |
| $\rho_1$ | 01000 | 11110 | 01111 | 01100 | 00100 |
| $\rho_2$ | 00100 | 01111 | 11110 | 01100 | 01000 |
| $\rho_3$ | 00010 | 01100 | 01100 | 10011 | 00010 |
| $\rho_4$ | 00001 | 00100 | 01000 | 00010 | 10000 |

Full Bispectrum $\{\beta_{\rho_i,\rho_j}\}_{i,j=0,..,4}$
Selective Bispectrum $\{\beta_{\rho_0,\rho_0}, \beta_{\rho_4,\rho_0}, \beta_{\rho_4,\rho_4}\}$

Full Bispectrum $\{\beta_{\rho_i,\rho_j}\}_{i,j=0,..,4}$
Selective Bispectrum $\{\beta_{\rho_0,\rho_0}, \beta_{\rho_1,\rho_0}, \beta_{\rho_1,\rho_1}, \beta_{\rho_1,\rho_2},\}$

Figure 3: Comparison of full and selective $G$-Bispectra for the dihedral group $D_4$ (left) and the octahedral group $O_h$ (right). The Kronecker tables of both groups show which irreps emerge from the decomposition into irreps of the tensor product of irreps $\rho_i \otimes \rho_j$. The colored boxes highlight the bispectral coefficients chosen for the full and selective Bispectra. Our proposed selective Bispectrum captures the same information as the full Bispectrum but with significantly fewer coefficients.

**New Theorems** From now on, we assume that the Fourier transform $\mathcal{F}(\Theta)$ only features *non-zero elements*, or *invertible matrices* in the case of non-scalar Fourier coefficients. This assumption is supported by the zero probability of encountering this corner case (an arbitrarily small perturbation of any signal makes this assumption true).

We first extend the above results to all commutative groups. The proof relies on the fact that every finite commutative group is the direct sum of finitely many cyclic groups.

**Theorem 4.3.** *For finite commutative groups $G$, the $G$-Bispectrum can be inverted using $|G|$ coefficients if $\mathcal{F}(\Theta)_\rho \neq 0$ for all $\rho \in G$.*

See Appendix D for the proof and derivation of the inversion for the specific case of commutative groups. We note that our approach to inversion is symbolic, in that a solution can be expressed explicitly as a formula in terms of the input. Other approaches are also possible to determine an inverse, such as using least squares [11] or more recent spectral methods [4].

We now extend the result to dihedral groups. Dihedral groups are ubiquitous in signal processing and deep learning because they represent the group of rotations and reflections.

**Theorem 4.4.** *For any dihedral group $D_n$ (symmetries of the $n$-gon), $n \in \mathbb{N}_0$, we need at most $\lfloor \frac{n-1}{2} \rfloor + 2$ bispectral matrix coefficients for inversion if $\det(\mathcal{F}(\Theta)_\rho) \neq 0$ for all irreps $\rho$ of $D_n$. This corresponds to $1 + 4 + 16 \cdot \lfloor \frac{n-1}{2} \rfloor \approx 4|D_n|$ scalar values.*

The proof is provided in Appendix E. We now extend the result to octahedral and full octahedral groups, that are related to the symmetries of the octahedron. These groups are very important in signal processing and deep learning of 3D images.

**Theorem 4.5.** *For the octahedral group $O$ which has $|G| = 24$ group elements and $5$ irreps, we need only $4$ $G$-Bispectral coefficients in the selective $G$-Bispectrum. this corresponds to $172$ scalars. For the full octahedral group $FO$ which has $|G| = 48$ elements, we only need $6$ $G$-Bispectral coefficients in the selective $G$-Bispectrum to perform inversion. This corresponds to $334$ scalars.*

A sketch of proof is provided in Appendix F given the redundancy of the procedure. We see that the selective $G$-Bispectrum uses only $4$ coefficients, compared to $25$ coefficients needed for the full $G$-Bispectrum of the octahedral group. For the full octahedral group, it requires only $6$ coefficients compared to the $100$ coefficients of the full $G$-Bispectrum. In Figure 3, we compare the full and selective $G$-Bispectra of the dihedral group $D_4$ (symmetries of the square) and the octahedral group.

## 5 Experimental results

**Implementation and architecture** Our implementation of the selective $G$-bispectrum layer is based on the `gtc-invariance` repository, implementing the $G$-CNN with $G$-convolution and $G$-TC layer [28] and relying itself on the `escnn` library [3, 32]. The implementations related to this section can be found at the `g-invariance` repository.

| Invariance layer | Computational Complexity | Ouput size | Complete G-invariant |
|---|---|---|---|
| $G$-TC | $K\mathcal{O}(|G|^3)$ | $K\mathcal{O}(|G|^2)$ | ✓ |
| Full $G$-Bispectrum | $K\mathcal{O}(|G|^2)$ | $K\mathcal{O}(|G|^2)$ | ✓ |
| Select. $G$-Bispectrum | $K\mathcal{O}(|G|\log|G| \text{ or } |G|^2)$ | $K\mathcal{O}(|G|)$ | ✓ |
| Max $G$-pooling | $K\mathcal{O}(|G|)$ | $K\mathcal{O}(1)$ | ✗ |
| Avg $G$-pooling | $K\mathcal{O}(|G|)$ | $K\mathcal{O}(1)$ | ✗ |

Table 1: $G$-CNN invariant layers and their computational cost and output size. $K$ is the number of filters. The selective $G$-Bispectrum that we propose is the complete $G$-invariant layer with the lowest time and space complexity. It reduces significantly the cost compared to the $G$-TC layer while preserving its completeness.

We propose an experimental assessment of the newly proposed selective $G$-Bispectrum layer by comparing it with the Avg $G$-pooling, the Max $G$-pooling, the $G$-TC as invariance operations after the $G$-convolution of a $G$-CNN on the classification problems of the MNIST dataset of handwritten digits [23], the EMNIST dataset of handwritten letters [5] with standard train-test division. These datasets count 10 and 26 classes, respectively. We obtain transformed versions of the datasets – $G$-MNIST/EMNIST – by applying a random action $g \in G$ on each image in the original dataset.

The objective of our experiments is to isolate the speed-up of the $G$-Bispectrum layer. Hence, we consider architectures that only differ by the invariant layer in the classification task, following the experimental set up by [28]. The neural network architecture is composed of a $G$-convolution, a $G$-invariant layer, and finally a Multi-Layer-Perceptron (MLP), itself composed of three fully connected layers with ReLU nonlinearity. Finally, a fully connected linear layer is added to perform classification. The MLP's widths are tuned to match the number of parameters across each neural network model. The details are given in Appendix G. We highlight here that the pursued objective is to compare the differences in performances of the $G$-invariant layers, not to provide the state-of-the-art accuracy on the datasets involved. Henceforth, we do not optimize the architectures to reach the highest possible accuracy. We set simple architectures providing interpretable results for analysis. The experiments a performed using 8 cores of a NVIDIA A30 GPU.

**Training speed performance**   Table 1 recalls the theoretical complexities of the different layers. The computational cost of computing the selective $G$-Bispectrum is $\mathcal{O}(|G|\log|G|)$ if an FFT algorithm is available on $G$ [9], and $\mathcal{O}(|G|^2)$ with classical DFT. in Figure 4, we report the average training times on SO(2)/O(2)-MNIST for 10 runs as the discretization $C_n/D_n$ of SO(2)/O(2) varies. In the first case, we use the FFT and observe that the Max $G$-pooling and $G$-Bispectrum training time scale linearly whereas it scales quadratically for the $G$-TC. For O(2), we perform a classic DFT on $D_n$ so that the $G$-Bispectrum scales worth. However, an FFT could be implemented to speed-up the process.

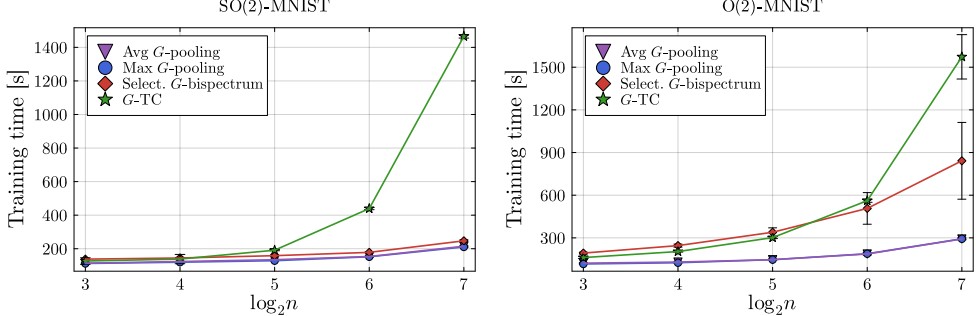

Figure 4: Evolution of the average training times for the different invariant layers. The parameter $n$ is the size of the groups $C_n$ and $D_n$. The average and standard deviations are obtained over 10 runs. For all runs, the number of parameters of the complete neural network (filters and MLP) is set to 50000 and 150000 for SO(2) and O(2) respectively. Standard deviations are reported by vertical intervals. When a FFT is available, our selective G-Bispectrum significantly outperforms other complete G-invariant pooling layers in terms of speed. Specifically, when working with $C_{2^7}$, training on a dataset of 60000 images takes only 247 seconds, whereas the $G$-TC requires 1465 seconds.

| Dataset | Group | $G$ | Pooling | $K$ filters | Avg acc. | Std. dev. | Param. count |
|---|---|---|---|---|---|---|---|
| MNIST | SO(2) | $C_8$ | Avg $G$-pooling | 24 | 0.74 | < 0.01 | 50247 |
| | | | Max $G$-pooling | 24 | 0.96 | < 0.01 | 50247 |
| | | | Select. $G$-Bispectrum | 24 | 0.95 | < 0.01 | 49116 |
| | | | $G$-TC | 24 | 0.96 | < 0.01 | 48385 |
| | O(2) | $D_8$ | Avg $G$-pooling | 4 | 0.60 | < 0.01 | 147675 |
| | | | Max $G$-pooling | 4 | 0.78 | < 0.01 | 147675 |
| | | | Select. $G$-Bispectrum | 4 | 0.93 | < 0.01 | 143029 |
| | | | $G$-TC | 4 | 0.96 | < 0.01 | 142220 |
| EMNIST | SO(2) | $C_8$ | Avg $G$-pooling | 24 | 0.40 | < 0.01 | 50195 |
| | | | Max $G$-pooling | 24 | 0.76 | < 0.01 | 50195 |
| | | | Select. $G$-Bispectrum | 24 | 0.77 | < 0.01 | 49254 |
| | | | $G$-TC | 24 | 0.80 | < 0.01 | 48494 |
| | O(2) | $D_8$ | Avg $G$-pooling | 20 | 0.38 | < 0.01 | 48832 |
| | | | Max $G$-pooling | 20 | 0.71 | < 0.01 | 48832 |
| | | | Select. $G$-Bispectrum | 20 | 0.74 | < 0.01 | 47320 |
| | | | $G$-TC | 20 | 0.79 | < 0.01 | 46954 |

Table 2: Results of numerical experiments averaged over 10 runs with Avg $G$-pooling, Max $G$-pooling, our selective $G$-Bispectrum and $G$-TC. The experiments are performed on SO(2)/O(2)-MNIST and SO(2)/O(2)-EMNIST. The table shows the number of filters, the average classification accuracy, standard deviation and parameter count. This table shows that the selective $G$-Bispectrum conserves the accuracy of the $G$-TC at an equivalent number of parameters.

**Classification Performance**   We compare the performances of the $G$-Bispectrum layer with respect to the $G$-TC, the Max $G$-pooling and the Avg $G$-pooling models, trained on the SO(2)/O(2)-MNIST/EMNIST datasets and we assess the accuracy by averaging the validation accuracy over 10 runs. The classification accuracy is provided in Table 2. For the experiments in Table 2, the following pattern holds: at equivalent number of parameters, the more computationally expensive the pooling layer, the better the accuracy. However, the use of the $G$-TC becomes prohibitive when $|G|$ increases. In the next section, we discuss the settings where each invariant layer should be preferred, and highlight each invariant layer's strengths and weaknesses.

**Discussion on the choice of invariant layer**   The first observation from Table 2 is though the *selective* $G$-Bispectrum is complete, the model obtains slightly lower accuracy than $G$-TC. This observation might be surprising at first, since we prove mathematically in Section 4 that the selective $G$-Bispectrum is complete, just as the full version. An explanation to this lies in the paradoxes of the Universal Approximation Theorem [13]. Just because an arbitrarily large MLP can theoretically fit any function, this does not imply that it will happen for a practical, limited MLP. In practice, we hypothesize that the redundancy of the $G$-TC allows the MLP to distinguish inputs more easily. If the size of the model allows it, the $G$-TC or the full $G$-Bispectrum will provide better accuracy. However, when the size of the group is big, their use is often out of reach while the selective $G$-Bispectrum is scalable. In Table 2, we also notice that the Max $G$-pooling performs well compared to the others even though it is not complete. This is because we have many filters that allow for refined classification. Indeed, assume $f, \phi_k$ are black-and-white images with $N$ pixels. In consequence, $\max_g \Theta_k(g) \in \{0, 1, ..., N\}$ for $k = 1, 2, ..., K$. The Max $G$-pooling allows a maximum separation of $(N + 1)^K$ classes. In practice, this value is not reached, but it explains why Max $G$-pooling performs well. Figure 5 highlights this dependency of the Max $G$-pooling on the number of filters since the accuracy drops to less than $60\%$ with 2 filters. In comparison, the $G$-TC and the selective $G$-Bispectrum, which are *complete*, keep an accuracy above $85\%$ with 2 filters.

**Completeness**   To conclude our numerical experiments, we study the robustness of the selective $G$-Bispectrum to adversarial attacks, following the analysis in Sanborn & Miolane [28, Figure 2]. Given an image $\widehat{f} : X \mapsto \mathbb{R}^c$ and a filter $\phi : X \mapsto \mathbb{R}^c$, they numerically verified the robustness (=completeness) of the $G$-TC by showing that

$$f^* \in \arg\min_{f : X \mapsto \mathbb{R}^c} \|T(\phi * f) - T(\phi * \widehat{f})\|_2^2 \iff f^* = \alpha(g, \widehat{f}) \text{ for some } g \in G. \tag{3}$$

Indeed, Sanborn & Miolane [28, Figure 2] shows that only images that are identical up to rotation/reflection can yield the same $C_n/D_n$-TC. That is, the $G$-CNN with $G$-TC can not be "fooled"

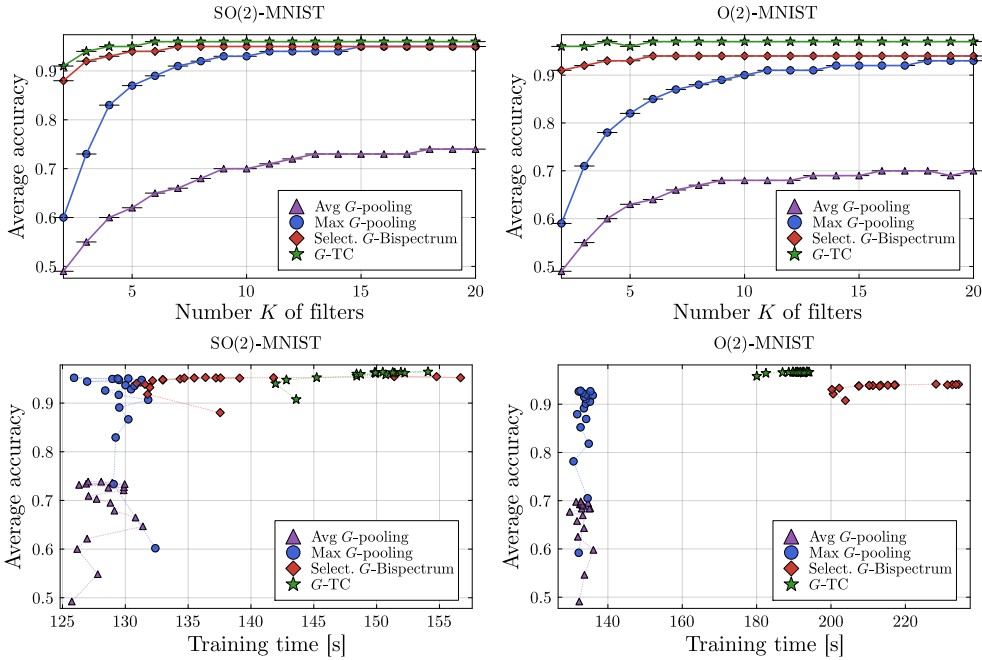

Figure 5: At the top: Evolution of the average classification accuracy with rotated MNIST (SO(2)-MNIST) and rotated-reflected MNIST (O(2)-MNIST) over 10 runs when the number of filters varies from 2 to 20 for the Avg $G$-pooling, the Max $G$-pooling, the selective $G$-Bispectrum and the $G$-TC. The number of parameters of each model is maintained equal for fair comparison. The standard deviations are represented using vertical intervals. With the selective $G$-Bispectrum layer, we can reduce the number of convolutional filters needed for a given accuracy. For example, with only $K = 2$ filters, we achieve 96% accuracy, compared to 63% with the Max $G$-pooling layer. Our approach allows $G$-CNNs to maintain competitive accuracy while using smaller neural networks. At the bottom, the same results are displayed with time instead of the number of filters on the $x$-axis. The dotted lines reproduce the evolution of $K$ from the figures at the top. We can observe that the selective $G$-Bispectrum is faster than the $G$-TC when a FFT is available, thus here in the case of SO(2)-MNIST. Recall that an FFT can be implemented for many groups [9]

since only input in the same orbit yield the same output. We perform a similar experiment in Figure 6. Moreover, it is well-known that the $G$-convolution $\phi * f$ is $G$-equivariant. Hence, an equivalent experiment is to show that

$$\Theta^* \in \arg \min_{\Theta : G \mapsto \mathbb{R}} \|T(\Theta) - T(\widehat{\Theta})\|_2^2 \iff \Theta^* = \alpha(g, \widehat{\Theta}) \text{ for some } g \in G.$$

In Figure 7, we show that the selective $G$-Bispectrum $\beta_{sel}$ is robust to adversarial attacks by solving

$$\Theta^* \in \arg \min_{\Theta : G \mapsto \mathbb{R}} \|\beta_{sel}(\Theta) - \beta_{sel}(\widehat{\Theta})\|_2^2. \tag{4}$$

The signals are indeed recovered up to a translation, i.e., a group action of $C_{30}$. Moreover, despite (4) only optimizes using the selective $G$-Bispectrum, the full $G$-Bispectrum is correctly recovered. This is an additional evidence of the completeness of the selective $G$-Bispectrum.

## 6 Conclusion and Future works

In this paper, we introduced a new type of complete invariant layer for $G$-invariant CNNs – called *selective $G$-Bispectrum layer* – with the objective of increasing the accuracy and robustness of $G$-CNNs compared to those implemented with the initially proposed Max $G$-pooling. The $G$-TC layer also achieves this goal, but at an output cost of $\mathcal{O}(|G|^2)$ coefficients and $\mathcal{O}(|G|^3)$ flops that prevents its application to large groups, while the selective $G$-Bispectrum layer only outputs $\mathcal{O}(|G|)$ coefficients. Building on the result of Kakarala [15] for cyclic groups, we have shown that the completeness of the *selective $G$-Bispectrum layer* holds for all commutative groups, all dihedral groups, the octahedral and full octahedral groups. In a suite of experiments, we provided a global

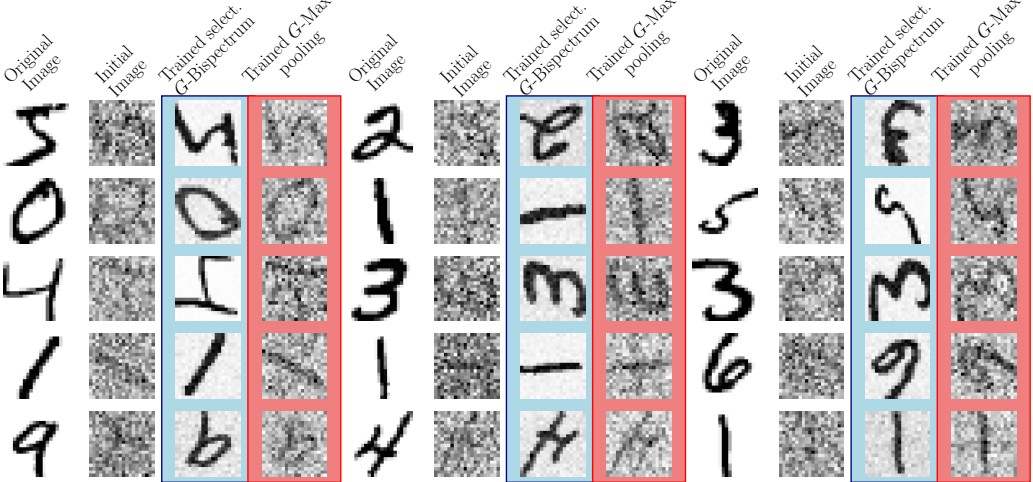

Figure 6: Adversarial attacks experiments with $G = C_4$. Images are optimized to output, respectively, a target selective $G$-bispectrum and a target $G$-Max pooling, obtained from an original image. The initial image is the initialization of the optimization process (3). After training, only for the selective $G$-Bispectrum (in blue), the recovered image is a copy of the original image up to group action (rotation). This is a numerical illustration of the robustness of the selective $G$-Bispectrum to adversarial attacks: one can not obtain the same output with an input that is not in the same class. On the other hand, $G$-Max pooling (in red) outputs a noisy image because it is not complete.

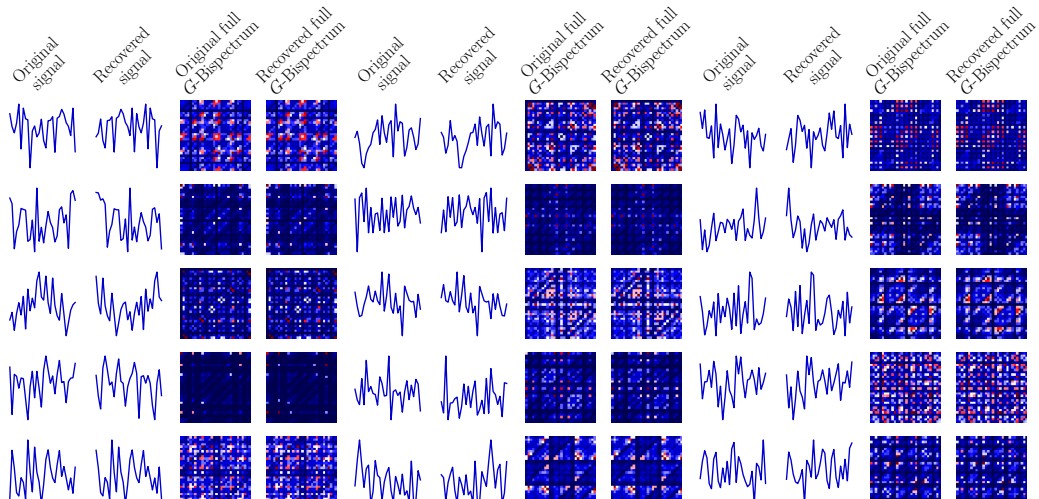

Figure 7: Numerical experiment of signal recovering from original signals $\{\widehat{\Theta}_i\}_{i=1}^{15}$ where $\widehat{\Theta}_i : G \mapsto \mathbb{R}$ by solving (4) with $G = C_{30}$. We use the gradient method with Armijo line search to solve (4). The recovered solutions $\{\Theta_i^*\}_{i=1}^{15}$ are represented and correspond to translations of the original signals. The moduli of the full $G$-Bispectra are also represented and are identical. This experiment corroborates the completeness of the selective $G$-Bispectrum since we are able to recover an unknown signal only from the knowledge of its selective $G$-Bispectrum.

picture of the strength and weaknesses of each invariant layer. We studied the performance in terms of training speed, classification accuracy and robustness to adversarial attacks. As a result, the selective $G$-Bispectrum paves the way for the development of complete invariant pooling layers that can accommodate larger group sizes and, hence, a larger set of symmetries.

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

# A   Background on groups

We introduce the fundamentals of group theory, which provide the foundation for the theory of $G$-CNNs. These notions can be found in [30].

**Definition A.1.** A *group* is a pair $(G, \cdot)$ where $G$ is a set and $\cdot : G \times G \mapsto G$ is an associative multiplication such that there is an identity element $e \in G$ (i.e., for all $g \in G$, $e \cdot g = g \cdot e = e$) and, for all $g \in G$, there is an inverse $g^{-1} \in G$ such that $g^{-1} \cdot g = g \cdot g^{-1} = e$.

A group is thus a set $G$ combined with a product $\cdot$ preserving the characteristics of $G$. Here, the established term "product" can be misleading. It denotes any operation which makes Definition A.1 true given the set $G$. For instance, $(\mathbb{R}, +)$ is a group. Another example is $\mathrm{GL}(\mathbb{R}^n)$, the $n \times n$ real invertible matrices, associated to the usual matrix product. This group is said to be *non-commutative* since $A \cdot B \neq B \cdot A$ in general for $A, B \in \mathrm{GL}(\mathbb{R}^n)$. An important group for us is the set $\{0, 1, ..., n-1\}$ associated with addition modulo $n$. It is usually written $\mathbb{Z}/n\mathbb{Z}$ and called the *cyclic group* $C_n$. A single group can arise in different contexts under seemingly distinct forms. For instance, $\mathbb{Z}/4\mathbb{Z}$ and the rotations leaving the square unchanged in $\mathbb{R}^2$ are fundamentally the same object. This observation gives rise to *representation theory*, a branch of group theory studying how the same abstract idea of a group can emerge under different forms.

**Definition A.2.** A *representation* of a group $(G, \cdot)$ is a pair $(\rho, V)$ where $V$ is a vector space and $\rho : G \mapsto \mathrm{GL}(V)$ is a group homomorphism, i.e., for all $g, h \in G$, $\rho(g \cdot h) = \rho(g)\rho(h)$. If $V$ is equipped with an inner product and if for all $g \in G$ and all $u, v \in V$, $\langle \rho(g)v, \rho(g)w \rangle = \langle u, v \rangle$, $\rho$ is *unitary*.

*Remark* A.3. Throughout this paper, we use the shorthand $G$ to refer to the group $(G, \cdot)$ and $\rho$ to refer to a representation $(\rho, V)$.

To illustrate Definition A.2, a representation of $C_n$ is given by the complex roots of unity, $\rho(k) = \exp\left(\frac{2\pi i}{n}k\right)$, on the complex one-dimensional vector space $V = \mathbb{C}$. Every group also admit the *trivial* representation: $\rho_0(g) = 1$ for all $g \in G$. There is a specific subset of these representations called the irreducible representations, *irreps* for short, being those that can not be expressed in a more compact form. The irreps are fundamental objects of group theory since they allow us to define an invertible Fourier transform on finite groups. The irreps are therefore needed to define the $G$-Bispectrum – i.e., the Fourier transform of the $G$-TC. The notion of irreps is derived from that of a $G$-invariant subspace, which we recall in Definition A.4.

**Definition A.4.** Given a representation $(\rho, V)$, a subspace $W \subseteq V$ is $G$-invariant if $\rho(g)w \in W$ for all $g \in G$, $w \in W$.

The formal definition of the irreps is then stated as the representations with no non-trivial invariant subspace.

**Definition A.5.** A non-zero representation $(\rho, V)$ of a group $G$ is irreducible if the only $G$-invariant subspaces of $V$ are $\{0\}$ and $V$ itself.

A single group acting on different spaces will have different representations. However, one can reveal the similarity between these representations by the mean of an equivalence relation.

**Definition A.6.** Two representations $(\rho, V)$ and $(\varphi, W)$ are *equivalent* if there exists an isomorphism $T : V \mapsto W$ such that for all $g \in G$, $\rho(g)T = T\varphi(g)$.

For the interested reader, the invertibility property of the Fourier transform is a consequence of the concepts of Pontryagin duality (commutative groups) and Tannaka-Krein duality (non-commutative groups); see, e.g., [14]. Our proofs will also rely on the notion of generating set of $G$, which we introduce here.

**Definition A.7.** A *generating set* $S$ of a group $(G, \cdot)$ is a subset $S \subset G$ such that every $g \in G$ can be expressed as a finite combination of the elements in $S$ and their inverses under the group action $\cdot$.

*Remark* A.8. It can be shown that every group $G$ of size $|G|$ has a generating set of size at most $\log_2 |G|$.

**Group Actions**   A group $(G, \cdot)$ represents a set of transformations such as rotations that can act on data such as images. We define formally how groups can indeed transform datasets through the concept of group action.

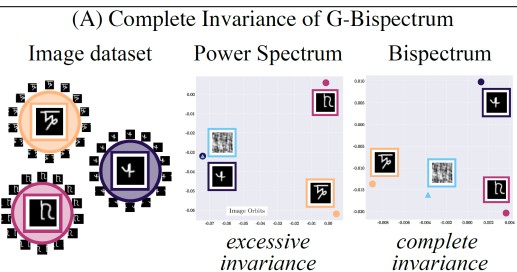

Figure 8: Illustration of the concepts of excessive and complete invariance to a group action. With the excessive invariance, samples from different classes can be mapped to the same output.

**Definition A.9.** Given a group $(G, \cdot)$, a *group action* $\alpha : G \times X \mapsto X$ is a function satisfying i) Identity: $\alpha(e, x) = x$, ii) Compatibility: $\alpha(h, \alpha(g, x)) = \alpha(h \cdot g, x)$ for any $x \in X$ and $h, g \in G$ and where $e$ is the identity of $G$.

Processing operations and neural networks can be designed so that they respect group actions: specifically, a group acting on the input (e.g., rotating an input image) should yield a group action on the output (e.g., a rotation of the output feature map). This is the notion of *G-equivariance*.

**Definition A.10.** A function $\psi : X \mapsto Y$ is *G-equivariant* if $\psi(\alpha_1(g, x)) = \alpha_2(g, \psi(x))$ for all $x \in X$ and all $g \in G$, where $\alpha_1$ and $\alpha_2$ are group actions on $X$ and $Y$, respectively.

For example, the $G$-convolution layer is $G$-equivariant by design [6]. An important problem in signal processing and deep learning is to achieve *invariance* to nuisance factors not relevant for the task. Many of these factors are describable as group actions (e.g. rotations, translations, scaling). Thus, we want processing methods and machine learning models to be $G$-invariant:

**Definition A.11.** A function $\psi : X \mapsto Y$ is *G-invariant* if $\psi(\alpha(g, x)) = \psi(x)$ for all $x \in X$ and all $g \in G$.

For example, the Max $G$-pooling $(\max_{g \in G} \Theta(g))$ traditionally follows a $G$-convolutional layer to remove the equivariance of the convolution and achieve $G$-invariance. A $G$-CNN is a neural network that consists of $G$-convolutional layers and a pooling/invariance operation. The main applications of our proposed selective $G$-Bispectrum operation is to act as a $G$-invariant pooling layer, that can conveniently replace the classical Max $G$-Pooling layer of $G$-CNNs, as shown in the rest of the paper.

**Clebsh-Jordan matrices**  Given a group $(G, \cdot)$ and a family of unitary irreps $\{\rho_i\}_{i=0}^{|\text{Irreps}|-1}$, the Clebsh-Jordan matrix is analytically defined for each pair $\rho_1, \rho_2$ as:

$$(\rho_1 \otimes \rho_2)(g) = C_{\rho_1, \rho_2} \Big[ \bigoplus_{\rho \in \mathcal{R}} \rho(g) \Big] C_{\rho_1, \rho_2}^\dagger. \tag{5}$$

**$G$-Bispectrum for Commutative Finite Groups**  The computation of the $G$-Bispectrum simplifies for commutative groups compared to Theorem 2.3, as recalled below.

**Theorem A.12.** *[17] If $(G, \cdot)$ is a commutative group and $\Theta : G \to \mathbb{R}$, the $G$-Bispectrum can be computed as*

$$\beta(\Theta)_{\rho_1, \rho_2} = \mathcal{F}(\Theta)_{\rho_1} \mathcal{F}(\Theta)_{\rho_2} \mathcal{F}(\Theta)_{\rho_1 \otimes \rho_2}^\dagger. \tag{6}$$

For commutative groups, the $G$-bispectral coefficients are complex scalars [30].

# B  Indeterminacy of $G$-Bispectrum inversion problem

It is important to state precisely which information we can possibly retrieve from the $G$-Bispectrum. A consequence of the $G$-invariance of $\beta(\Theta)$ is that the $G$-Bispectrum inversion problem is ill-posed. Recall that $G$-invariance means that for all $h \in G$, we have $\beta(\alpha(h, \Theta))_{\rho_1, \rho_2} = \beta(\Theta)_{\rho_1, \rho_2}$. Given

a function $\Theta : G \mapsto \mathbb{R}$, a possible definition for the group action on $\Theta$ is given by $\alpha(h, \Theta(g)) = \Theta(h^{-1} \cdot g)$ for all $h \in G$ (see, e.g., [6]). Therefore, for all $h \in G$, we have

$$\mathcal{F}(\alpha(h, \Theta))_\rho = \sum_{g \in G} \alpha(h, \Theta(g))\rho(g)^\dagger$$

$$= \rho(hh^{-1}) \sum_{g \in G} \Theta(h^{-1}g)\rho(g)^\dagger$$

$$= \rho(h)\mathcal{F}(\Theta)_\rho,$$

which shows that the $G$-Fourier transform $\mathcal{F}(\Theta)$ is $G$-equivariant. In consequence, recovering $\mathcal{F}(\Theta)$ from $\beta(\Theta)$ can at best be done up to an unknown factor $\rho(h)$. Moreover, as explained in [15, 20], the indeterminacy is not limited to $\rho(h)$. Take for instance $C_n$. An indeterminacy factor $\rho_k(h) = \exp\left(\frac{2\pi i h k}{n}\right)$ corresponds to a translation of $h \in C_n$ of the signal, $\widetilde{\Theta}(g) = \Theta(g + h)$. [15] showed that $h$ is not restricted to $C_n = \mathbb{Z}/n\mathbb{Z}$: it may take any value in $[0, n]$. The Bispectrum is not only invariant to a discrete set of rotations, but to the continuous group of rotations $\mathrm{SO}(2)$. The factor can thus be written $\exp(i\varphi k)$ where $\varphi \in [0, 2\pi)$.

## C    Selective $G$-Bispectrum inversion: known results

From now on, we assume that the Fourier transform $\mathcal{F}(\Theta)$ only features **non-zero elements**, or **invertible matrices** in the case of non-scalar Fourier transform. This assumption is supported by the zero probability of encountering this corner case.

**Cyclic groups $C_n$**    We start with $(G, \cdot) = (\mathbb{Z}/n\mathbb{Z}, + \bmod n) =: C_n$. Recall that the irreps are given by $\rho_k(g) = \exp(\omega_k g)$ where $\omega_k := \frac{2\pi i k}{n}$ for $k \in \mathbb{Z}/n\mathbb{Z}$ (see, e.g., [30]).

**Theorem C.1.** *[17] For cyclic groups $C_n$, the $C_n$-Bispectrum can be inverted using $|G| = n$ coefficients if $\mathcal{F}(\Theta)_\rho \neq 0$ for all $\rho \in C_n$.*

*Proof.* The Fourier coefficient associated to the trivial representation $\mathcal{F}(\Theta)_{\rho_0}$, is uniquely determined and can be recovered from Theorem A.12 by identifying phase and modulus:

$$\beta(\Theta)_{\rho_0, \rho_0} = |\mathcal{F}(\Theta)_{\rho_0}|^3 \exp\left(i \arg(\mathcal{F}(\Theta)_{\rho_0})\right). \tag{7}$$

We proceed using Pontryagin duality: the irreps $\{\rho_k\}_{k=1}^n$, form a group $\widehat{G}$ themselves, with $\widehat{G} = G$. In the case of the cyclic group $C_n$, notice that for all $j, k \in \mathbb{Z}/n\mathbb{Z}$, $\rho_j \otimes \rho_k = \rho_{j+k}$. Leveraging (6), we can use $\beta(\Theta)_{\rho_0, \rho_1}$ to recover $\mathcal{F}(\Theta)_{\rho_1}$:

$$|\mathcal{F}(\Theta)_{\rho_1}|^2 = \frac{\beta(\Theta)_{\rho_0, \rho_1}}{\mathcal{F}(\Theta)_{\rho_0}}. \tag{8}$$

Equation (8) leaves an indeterminacy on the phase of $\mathcal{F}(\Theta)_{\rho_1}$. This corresponds to the indeterminacy factor $\exp(i\varphi)$, $\phi \in [0, 2\pi)$ of Appendix B. It is inherited from the $G$-invariance of $\beta(\Theta)$ (it is not injective, hence you cannot distinguish inputs that have the same $G$-Bispectrum). For now, let $\arg(\mathcal{F}(\Theta)_{\rho_1}) = 0$. The key to recover all the other Fourier coefficients is to notice that $S = \{1\}$ is a generating set of $\widehat{G} = \mathbb{Z}/n\mathbb{Z}$. Therefore, computing sequentially

$$\mathcal{F}(\Theta)_{\rho_{k+1}} = \left(\frac{\beta(\Theta)_{\rho_1, \rho_k}}{\mathcal{F}(\Theta)_{\rho_1}\mathcal{F}(\Theta)_{\rho_k}}\right)^\dagger, \tag{9}$$

for $k = 1, 2, ..., n - 2$ recovers completely $\mathcal{F}(\Theta)$. We are not done yet because the phase we fixed before is not a valid shift. A valid phase shift for $\mathcal{F}(\Theta)_{\rho_1}$ is such that the shift w.r.t the original signal has the form $\exp\left(\frac{2\pi i h}{n}\right)$ for $h \in \mathbb{N}$. This valid phase shift is easy to find. It is the unique $\varphi \in [0, \frac{2\pi}{n})$ such that, if we define $\mathcal{F}(\widetilde{\Theta})_{\rho_k} = \exp(\varphi k)\mathcal{F}(\Theta)_{\rho_k}$ for all $k \in \mathbb{Z}/n\mathbb{Z}$, then we have $\mathcal{F}^{-1}(\mathcal{F}(\widetilde{\Theta})) \in \mathbb{R}^n$ (i.e., with no imaginary part). Note that this is an explicit method to recover a valid phase while [17] relies on its existence without explicit method to find it. The method is summarized in Algorithm 2 and illustrated in Figure 9. In consequence only the following $G$-bispectral coefficients are needed for completeness: $\beta(\Theta)_{\rho_0, \rho_0}, \beta(\Theta)_{\rho_0, \rho_1}$ and $\beta(\Theta)_{\rho_1, \rho_k}$ for $k = 1, 2, ..., n-2$. This makes a total of $n = |G|$ coefficients. We summarize this result in Theorem C.1. $\qquad \square$

**Algorithm 2** Bispectrum inversion on $\mathbb{Z}/n\mathbb{Z}$ [17]

1: **Input**: $\beta(\Theta)_{\rho_0,\rho_0}, \beta(\Theta)_{\rho_0,\rho_1}$ and $\beta(\Theta)_{\rho_1,\rho_k}$ for $k = 1, 2, ..., n-2$.

2: Compute $|\mathcal{F}(\Theta)_{\rho_0}| = (\beta(\Theta)_{\rho_0,\rho_0})^{\frac{1}{3}}$ and $\arg(\mathcal{F}(\Theta)_{\rho_0}) = \arg(\beta(\Theta)_{\rho_0,\rho_0})$.

3: Compute $|\mathcal{F}(\Theta)_{\rho_1}| = \left(\frac{\beta(\Theta)_{\rho_0,\rho_1}}{\mathcal{F}(\Theta)_{\rho_0}}\right)^{\frac{1}{2}}$ and set $\arg(\mathcal{F}(\Theta)_{\rho_1}) = 0$.

4: **for** $k = 1, 2, ..., n-2$ **do**

5:     Compute $\mathcal{F}(\Theta)_{\rho_{k+1}} = \left(\frac{\beta(\Theta)_{\rho_1,\rho_k}}{\mathcal{F}(\Theta)_{\rho_1}\mathcal{F}(\Theta)_{\rho_k}}\right)^{\dagger}$.

6: **end for**

7: Find $\varphi \in [0, \frac{2\pi}{n})$ such that $\mathcal{F}^{-1}(\mathcal{F}(\widetilde{\Theta})) \in \mathbb{R}^n$ where $\mathcal{F}(\widetilde{\Theta})_{\rho_k} = \exp(\phi k)\mathcal{F}(\Theta)_{\rho_k}$.

8: **Return** $\mathcal{F}(\widetilde{\Theta})$ (up to group action).

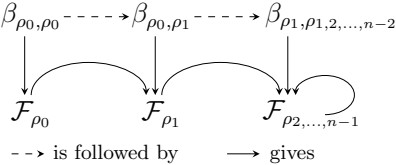

Figure 9: Illustration of Algorithm 2. The Bispectrum coefficients allow to recover the Fourier transform sequentially, up to group action. First, $\beta_{\rho_0,\rho_0}$ gives $\mathcal{F}_{\rho_0}$, which, combined with $\beta_{\rho_0,\rho_1}$ gives $\mathcal{F}_{\rho_1}$ etc.

## D    Selective $G$-Bispectrum inversion: commutative groups

Here, we prove the theorem stated in the main text and recalled below:

**Theorem D.1.** *For finite commutative groups $G$, the $G$-Bispectrum can be inverted using $|G|$ coefficients if $\mathcal{F}(\Theta)_\rho \neq 0$ for all $\rho \in G$.*

Specifically, we extend Algorithm 2 to all commutative groups, based on the Theorem D.2. That is, we design a method for the direct sum of finitely many cyclic groups.

**Theorem D.2.** *(see, e.g., [26]) Every finite commutative group $G$ is isomorphic to a finite direct sum of cyclic groups: $G \cong \bigoplus_{l=1}^{L} \mathbb{Z}/n_l\mathbb{Z}$ where $L \in \mathbb{N}$ and $n_l \in \mathbb{N}$ for $l = 1, 2, ..., L$.*

For all $\mathbf{k} \in G$ ($\mathbf{k}$ is integer-valued vector of length $L$), the irreps $\rho_{\mathbf{k}} : G \mapsto \mathbb{C}$ are given by $\rho_{\mathbf{k}}(g) = \prod_{l=1}^{L} \exp(\frac{2\pi i \mathbf{k}_l}{n_l} g_l)$. The number of irreps is $|G| = \prod_{l=1}^{L} n_l$. We detail and prove in Theorem D.3 the procedure to invert the $G$-Bispectrum on commutative groups. The procedure is summarized in Algorithm 3 where we use the two following notations.

1. $\mathbf{e}^l$ denotes the basis vector in $\mathbb{Z}^L$ such that $\mathbf{e}_k^l = 1$ if $k = l$ and $\mathbf{e}_k^l = 0$ otherwise.

2. $K^l := \{t\mathbf{e}^l + \mathbf{k} \mid t = 1, 2..., n_l - 1 \text{ and } \mathbf{k} \in K^{l-1}\}$ for $l = 1, 2, ..., L$.

The sets $K^l$ are a recursively constructed such that $K^L \cong G$. For $G = (\mathbb{Z}/3\mathbb{Z})^3$, the sets $K_1, K_2, K_3$ are represented in Figure 10.

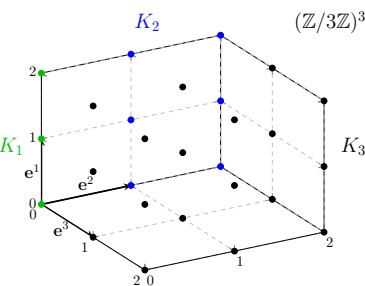

Figure 10: Representation of the sets $K_1, K_2$ and $K_3$ for $G = (\mathbb{Z}/3\mathbb{Z})^3$.

**Theorem D.3.** *For finite commutative groups $G$, the $G$-Bispectrum can be inverted using $|G|$ coefficients if $\mathcal{F}(\Theta)_\rho \neq 0$ for all $\rho \in G$.*

*Proof.* Notice that for the commutative groups, we keep the property $\rho_{\mathbf{k}} \otimes \rho_{\mathbf{k}'} = \rho_{\mathbf{k}+\mathbf{k}'}$ where $(\mathbf{k} + \mathbf{k}')_l := \mathbf{k}_l + \mathbf{k}'_l \mod n_l$ for $l = 1, 2, ..., L$. The first step is to obtain a generating set $S$ of the

irreps is of size $L$. It is given by the usual basis vectors $S = \{\mathbf{e}^l \in \mathbb{Z}^L \text{ for } l = 1, ..., L\}$ where

$$\mathbf{e}^l_k = \begin{cases} 1 \text{ if } k = l, \\ 0 \text{ otherwise.} \end{cases}$$

By Theorem A.12, it is sufficient to have the Fourier coefficients associated to each generating element in $S$ to recover all the Fourier coefficients. Indeed, knowing $\mathcal{F}(\Theta)_{\rho_{\mathbf{k}_1}}$ and $\mathcal{F}(\Theta)_{\rho_{\mathbf{k}_2}}$ allows us to compute $\mathcal{F}(\Theta)_{\rho_{\mathbf{k}_1 + \mathbf{k}_2}}$. By definition of the generating set $S$, we can thus recover $\mathcal{F}(\Theta)_{\rho_{\mathbf{k}}}$ for all $\mathbf{k} \in G$. The moduli of the coefficients $\mathcal{F}(\Theta)_{\rho_{\mathbf{e}^l}}$ for $l = 1, 2, ..., L$ can be computed as follows:

$$|\mathcal{F}(\Theta)_{\rho_{\mathbf{e}^l}}| = \left( \frac{\beta(\Theta)_{\rho_{\mathbf{0}}, \rho_{\mathbf{e}^l}}}{\mathcal{F}(\Theta)_{\rho_{\mathbf{0}}}} \right)^{\frac{1}{2}}, \tag{10}$$

where $\mathcal{F}(\Theta)_{\rho_{\mathbf{0}}}$ is the Fourier coefficient of the trivial representation (computed as in (7)). We claim that the phase can be fixed independently for each label $l$, thus $L$ times. This is because only one factor $h_l$ remains among the $L$ independent factors in $\mathbf{h}$:

$$\mathcal{F}(\alpha(\mathbf{h}, \Theta))_{\rho_{\mathbf{e}^l}} = \rho_{\mathbf{e}^l}(\mathbf{h}) \mathcal{F}(\Theta)_{\rho_{\mathbf{e}^l}} = \exp\left( \frac{2\pi i h_l}{n_l} \right) \mathcal{F}(\Theta)_{\rho_{\mathbf{e}^l}},$$

for all $l = 1, 2, ., L$. Therefore, fixing an arbitrary phase in (10) only fixes $h_l$. Again, the indeterminacy factor $h_l$ is not restricted to $\mathbb{Z}/n_l\mathbb{Z}$ but can belong to $[0, n_l]$. We will have to solve this issue further. For now, we set the phase of $\mathcal{F}(\Theta)_{\rho_{\mathbf{e}^l}}$ to zero. Once the Fourier coefficients are known for all the generators of the group of the irreps $\{\rho_{\mathbf{e}^l}\}_{l=1}^L$, it remains to combine them to obtain all the elements in the groups and, consequently, all the associated Fourier coefficients.

At this point, it helps to consider the problem geometrically. Each irreps $\rho_{\mathbf{k}}$ can be associated to its integer coordinate $\mathbf{k}$ inside a hyper-rectangle in $\mathbb{R}^L$, whose length of edges is $n_l$ for $l = 1, 2, ..., L$. We combine the coordinates $\mathbf{e}^l$ to obtain all the possible integer coordinates inside the hyper-rectangle. First, we can obtain the $L$ orthogonal edges of the hyper-rectangle. For $l = 1, 2, ..., L$, $\rho_{\mathbf{e}^l} \otimes \rho_{\mathbf{e}^l}$ gives $\rho_{2\mathbf{e}^l}$, $\rho_{\mathbf{e}^l} \otimes \rho_{2\mathbf{e}^l}$ gives $\rho_{3\mathbf{e}^l}$, etc. This is in fact the procedure of Algorithm 1. Now, we combine the edges to generate the inside of the hyper-rectangle. We proceed iteratively. For $l = 1, 2, ..., L$, we define $K^l := \{t\mathbf{e}^l + \mathbf{k} \mid t = 1, 2, .., n_l - 1 \text{ and } \mathbf{k} \in K^{l-1}\}$ and $K^0 := \{\mathbf{0}\}$. This construction is such that $K^l \cong \bigoplus_{j=1}^l \mathbb{Z}/n_j\mathbb{Z}$ and $K^L = G$. We generate the missing Fourier coefficients by combining the ones associated to the generating set of $G$. For $l = 2, ..., L$, compute

$$\mathcal{F}(\Theta)_{\rho_{t\mathbf{e}^l + \mathbf{k}}} = \left( \frac{\beta(\Theta)_{\rho_{t\mathbf{e}^l}, \rho_{\mathbf{k}}}}{\mathcal{F}(\Theta)_{\rho_{t\mathbf{e}}} \mathcal{F}(\Theta)_{\rho_{\mathbf{k}}}} \right)^{\dagger}, \tag{11}$$

for $t = 1, 2, ..., n_l - 1$, for all $\mathbf{k} \in K^{l-1}$. Intuitively for $L = 3$, we obtain first an edge, then a face and finally the full parallelepiped. To conclude, we reproduce the procedure from Algorithm 2 to find a valid phase shift $\varphi_l$ in each basis direction $\mathbf{e}^l$. The last step is then to compute, for all $\mathbf{k} \in G$,

$$\mathcal{F}(\widetilde{\Theta})_{\rho_{\mathbf{k}}} = \mathcal{F}(\Theta)_{\rho_{\mathbf{k}}} \prod_{l=1}^L \exp(\phi_l k_l).$$

The procedure is summarized in Algorithm 3 and illustrated in Figure 11. It shows that the bispectral coefficients needed for completeness are: $\beta(\Theta)_{\rho_0, \rho_0}$, $\beta(\Theta)_{\rho_0, t\rho_{\mathbf{e}}^l}$, $\beta(\Theta)_{t\rho_{\mathbf{e}}^l, \rho_{\mathbf{k}}}$ for $l = 1, 2, ..., L$, $t = 1, 2, ..., n_l - 2$ and all $\mathbf{k} \in K^{l-1}$. We recover exactly one Fourier coefficient per $G$-bispectrum coefficient. This makes thus a total of $|G|$ bispectral coefficients precisely and proves the following theorem. $\qquad\square$

## E  Selective $G$-Bispectrum inversion: dihedral groups

**Dihedral group $D_n$**    The dihedral group $D_n$ is the group of all symmetries of the $n$-gon. Mathematically, it is defined as

$$D_n := \langle x, a \mid a^n = x^2 = e, \; xax = a^{-1} \rangle, \tag{12}$$

---

**Algorithm 3** Bispectrum inversion for finite commutative groups ($G = \bigoplus_{l=1}^{L} \mathbb{Z}/n_l\mathbb{Z}$).

---

1: **Input**: $|G|$ bispectral coeffs.
2: Compute $|\mathcal{F}(\Theta)_{\rho_0}| = (\beta(\Theta)_{\rho_0,\rho_0})^{\frac{1}{3}}$ and $\arg(\mathcal{F}(\Theta)_{\rho_0}) = \arg(\beta(\Theta)_{\rho_0,\rho_0})$.
3: **for** $l = 1, ..., L$ **do**
4:     Compute $|\mathcal{F}(\Theta)_{\rho_{\mathbf{e}^l}}| = \left( \frac{\beta(\Theta)_{\rho_0,\rho_{\mathbf{e}^l}}}{\mathcal{F}(\Theta)_{\rho_0}} \right)^{\frac{1}{2}}$ and set $\arg\left(\mathcal{F}(\Theta)_{\rho_{\mathbf{e}^l}}\right) = 0$.
5:     **for** $t = 1, ..., n_l - 1$ **do**
6:         **if** $t > 1$ **then**
7:             $F(\Theta)_{\rho_{t\mathbf{e}^l}} = \left( \frac{\beta(\Theta)_{\rho_{\mathbf{e}^l},\rho_{(t-1)\mathbf{e}^l}}}{\mathcal{F}(\Theta)_{\rho_{\mathbf{e}^l}}\mathcal{F}(\Theta)_{\rho_{(t-1)\mathbf{e}^l}}} \right)^{\dagger}.$
8:         **end if**
9:         **for** $\mathbf{k} \in K^{l-1} \setminus \{\mathbf{0}\}$ **do**
10:             Compute $\mathcal{F}(\Theta)_{\rho_{t\mathbf{e}^l + \mathbf{k}}} = \left( \frac{\beta(\Theta)_{\rho_{t\mathbf{e}^l},\rho_{\mathbf{k}}}}{\mathcal{F}(\Theta)_{\rho_{t\mathbf{e}^l}}\mathcal{F}(\Theta)_{\rho_{\mathbf{k}}}} \right)^{\dagger}.$
11:         **end for**
12:     **end for**
13: **end for**
14: **for** $l = 1, 2, ..., L$ **do**
15:     Find $\varphi_l \in [0, \frac{2\pi}{n})$ s.t. $\mathcal{F}^{-1}(\mathcal{F}(\widetilde{\Theta})_{\rho_{(1,..,n_l)\mathbf{e}^l}}) \in \mathbb{R}^{n_l}$
        where $\mathcal{F}(\widetilde{\Theta})_{\rho_{k\mathbf{e}^l}} = \exp(\phi_l k)\mathcal{F}(\Theta)_{\rho_{k\mathbf{e}^l}}.$
16: **end for**
17: **for** $\mathbf{k} \in G$ **do**
18:     $\mathcal{F}(\widetilde{\Theta})_{\rho_{\mathbf{k}}} = \mathcal{F}(\Theta)_{\rho_{\mathbf{k}}} \prod_{l=1}^{L} \exp(\phi_l k_l).$
19: **end for**
20: **Return** $\mathcal{F}(\widetilde{\Theta})$ (up to group action).

---

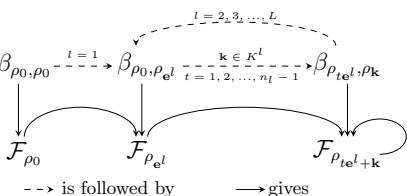

Figure 11: Illustration of Algorithm 3. The Bispectrum coefficients allow to recover the Fourier transform sequentially, up to group action.

where $a$ is the *rotation* and $x$ is the *reflection*, and they form a generating set of $D_n$. We will only consider the case $n > 2$ since the cases $n = 1$ and $n = 2$ are commutative groups covered by the previous subsection, while $n > 2$ gives non-commutative groups. The 2D irreps of $D_n$ are given by

$$\rho_k(a^l x^m) = \begin{bmatrix} \cos(\omega_l k) & -\sin(\omega_l k) \\ \sin(\omega_l k) & \cos(\omega_l k) \end{bmatrix} \begin{bmatrix} 1 & 0 \\ 0 & -1 \end{bmatrix}^m, \tag{13}$$

where $\omega_l = \frac{2\pi l}{n}$ for $k = 1, 2, ..., \lfloor \frac{n-1}{2} \rfloor$. There are also 2 or 4 1D irreps if $n$ is odd or even, respectively. We denote these 1D irreps by $\rho_0$, $\rho_{01}$, $\rho_{02}$, and $\rho_{03}$ (see Appendix E.1). The two last ones only exist for $n$ even.

**Theorem E.1.** *For the family of dihedral groups $D_n$, we need at most $\lfloor \frac{n-1}{2} \rfloor + 2$ bispectral matrix coefficients for inversion if $\det(\mathcal{F}(\Theta)_\rho) \neq 0$ for all irreps $\rho$ of $D_n$. This corresponds to $1 + 4 + 16 \cdot \lfloor \frac{n-1}{2} \rfloor \approx 4|D_n|$ scalar values.*

*Proof.* In view of section C, we wish to show that there is an irrep $\rho_1$ that generates all the irreps of $D_n$. As in the cyclic and commutative cases, we can first deduce $\mathcal{F}(\Theta)_{\rho_0}$ from $\beta(\Theta)_{\rho_0,\rho_0}$. Now, we have

$$\mathcal{F}(\Theta)_{\rho_1}\mathcal{F}(\Theta)_{\rho_1}^{\dagger} = \frac{\beta(\Theta)_{\rho_0,\rho_1}}{\mathcal{F}(\Theta)_{\rho_0}}. \tag{14}$$

The novelty for this non-commutative group is that $\mathcal{F}(\Theta)_{\rho_1}$ is a $2 \times 2$ matrix. After computing the eigenvalue decomposition $\frac{\beta(\Theta)_{\rho_0,\rho_1}}{\mathcal{F}(\Theta)_{\rho_0}} = V\Lambda V^{\dagger}$, we can choose

$$\mathcal{F}(\Theta)_{\rho_1} = V\Lambda^{\frac{1}{2}}V^{\dagger}U. \tag{15}$$

for all unitary $U$ (i.e., $UU^{\dagger} = U^{\dagger}U = I$) to solve (14). In the case $\mathbb{Z}/n\mathbb{Z}$ the indeterminacy belonged to $SO(2)$, the continuous set of 2d rotations. For $D_n$, it belongs to $O(2)$, the continuous set of 2d

---

**Algorithm 4** Bispectrum inversion on the dihedral group $D_n$.

---

1: **Input**: $\beta(\Theta)_{\rho_0,\rho_0}$, $\beta(\Theta)_{\rho_0,\rho_1}$ and $\beta(\Theta)_{\rho_1,\rho_k}$ for $k = 1, 2, ..., \lfloor \frac{n-1}{2} \rfloor$.

2: Compute $|\mathcal{F}(\Theta)_{\rho_0}| = (\beta(\Theta)_{\rho_0,\rho_0})^{\frac{1}{3}}$ and $\arg(\mathcal{F}(\Theta)_{\rho_0}) = \arg(\beta(\Theta)_{\rho_0,\rho_0})$.

3: Compute $V\Lambda V^\dagger = \frac{\beta(\Theta)_{\rho_0,\rho_1}}{\mathcal{F}(\Theta)_{\rho_0}}$.

4: Set $\mathcal{F}(\Theta)_{\rho_1} = V\Lambda^{\frac{1}{2}}V^\dagger U$ with valid $U$ (existence of $U$ ensured but not computed easily).

5: **for** $k = 2, ..., \lfloor \frac{n-1}{2} \rfloor$ **do**

6:     Compute

$$\bigoplus_{\rho \in \rho_1 \otimes \rho_{k-1}} \mathcal{F}(\Theta)_\rho = \left( C_{\rho_1,\rho_{k-1}}^\dagger \left[ \mathcal{F}_{\rho_1} \otimes \mathcal{F}_{\rho_{k-1}} \right]^{-1} \beta_{\rho_1,\rho_2} C_{\rho_1,\rho_{k-1}} \right)^\dagger.$$

7: **end for**

8: **Return** $\mathcal{F}(\Theta)$ (up to group action).

---

rotations and reflections. For finite groups, Kakarala [15] ensures that the only choices for $U$ such that $\mathcal{F}(\Theta)$ is a Fourier transform on $G(= D_n)$ are such that $\Theta$ is identical to the original signal up to some group action $g \in D_n$. For $k = 2, ..., \lfloor \frac{n-1}{2} \rfloor$, we can then obtain

$$\bigoplus_{\rho \in \rho_1 \otimes \rho_{k-1}} \mathcal{F}(\Theta)_\rho = \left( C_{\rho_1,\rho_{k-1}}^\dagger \left[ \mathcal{F}(\Theta)_{\rho_1} \otimes \mathcal{F}(\Theta)_{\rho_{k-1}} \right]^{-1} \beta(\Theta)_{\rho_1,\rho_2} C_{\rho_1,\rho_{k-1}} \right)^\dagger. \tag{16}$$

It is shown in Appendix E.1.1 that $\rho_k$ appears in the tensor decomposition (5) of $\rho_1 \otimes \rho_{k-1}$ so that the for-loop can be applied. Moreover, we know from Appendix E.1.1 that $\rho_{01}$ appears in the decomposition of $\rho_1 \otimes \rho_1$ and, for $n$ even, $\rho_{02}, \rho_{03}$ in $\rho_1 \otimes \rho_{\frac{n}{2}-1}$. Thus the iteration recovers the complete DFT $\mathcal{F}(\Theta)$. $\beta(\Theta)_{\rho_0,\rho_0}$ is a scalar, $\beta(\Theta)_{\rho_0,\rho_1}$ is a $2 \times 2$ matrix and $\beta(\Theta)_{\rho_1,\rho_k}$, $k \neq 0$, is a $4 \times 4$ matrix. Hence the total number of scalars that is required is $1 + 4 + 16\lfloor \frac{n-1}{2} \rfloor$. $\qquad\square$

The procedure is summarized in Algorithm 4.

### E.1 The 1D irreps of $D_n$

We recall the definition of the dihedral group $D_n$ given in (12). The 1D irreps of $D_n$ can be found, e.g., in [30]. They are given by:

- $\rho_0(g) = 1$ for all $g \in D_n$.

- $\rho_{01}(g) = \begin{cases} 1 \text{ if } g \in \langle a \rangle, \\ -1 \text{ otherwise.} \end{cases}$

- If $n$ even, $\rho_{02}(g) = \begin{cases} 1 \text{ if } g \in \langle a^2, \, x \rangle, \\ -1 \text{ otherwise.} \end{cases}$

- If $n$ even, $\rho_{03}(g) = \begin{cases} 1 \text{ if } g \in \langle a^2, \, ax \rangle, \\ -1 \text{ otherwise.} \end{cases}$

To exemplify the 1D irreps, we give their values for $D_4$ in Table 3.

| $g$ | $e$ | $a$ | $a^2$ | $a^3$ | $x$ | $ax$ | $a^2x$ | $a^3x$ |
|---|---|---|---|---|---|---|---|---|
| $\rho_0(g)$ | 1 | 1 | 1 | 1 | 1 | 1 | 1 | 1 |
| $\rho_{01}(g)$ | 1 | 1 | 1 | 1 | -1 | -1 | -1 | -1 |
| $\rho_{02}(g)$ | 1 | -1 | 1 | -1 | 1 | -1 | 1 | -1 |
| $\rho_{03}(g)$ | 1 | -1 | 1 | -1 | -1 | 1 | -1 | 1 |

Table 3: 1D irreps of $D_4$.

#### E.1.1 Generation of the coefficients of $D_n$

Theorem E.1 makes two assertions that we verify explicitly in this appendix. First, it is said that $\rho_k$ is in the tensor decomposition (5) (TD) of $\rho_1 \otimes \rho_{k-1}$ for $k = 2, ..., \lfloor \frac{n-1}{2} \rfloor$ so that the iteration of

Algorithm 4 recovers all the Fourier coefficients associated to the 2D irreps. The second assertion to verify is that $\rho_{01}$ is in the TD of $\rho_1 \otimes \rho_1$ and, for $n$ even, $\rho_{02}, \rho_{03}$ in the TD of $\rho_1 \otimes \rho_{\frac{n}{2}-1}$ so that the Fourier coefficients associated to the 1D irreps are also recovered, asserting the validity of the inversion procedure. We provide an analytical proof of these two assertions. The proof is based on the theory of *character functions*.

**Definition E.2.** [30] Given a group $G$ and a representation $\rho$, the *character* of $\rho$ is the function $\chi_\rho : G \to \mathbb{R} : g \mapsto \mathrm{Tr}(\rho(g))$. $\chi_\rho$ is said to be an *irreducible* character if $\rho$ is an irreducible representation.

The character function $\chi_\rho$ is a *class function* on $G$, i.e., $\chi_\rho$ is constant on a conjugacy class of $G$. The space of class functions on a finite group $G$, written $\mathcal{S}_G$, can be equipped with an inner product $\langle \cdot, \cdot \rangle_G : \mathcal{S}_G \times \mathcal{S}_G \mapsto \mathbb{C}$ such that for $u, v \in \mathcal{S}_G$, we have

$$\langle u, v \rangle_G := \frac{1}{|G|} \sum_{g \in |G|} u(g)\overline{v(g)}. \tag{17}$$

The irreducible characters form an orthonormal basis w.r.t $\langle \cdot, \cdot \rangle_G$ for $\mathcal{S}_G$ [30], i.e.,

$$\langle \chi_{\rho_a}, \chi_{\rho_b} \rangle_G = \begin{cases} 1 \text{ if } \rho_a \sim \rho_b, \\ 0 \text{ otherwise.} \end{cases} \tag{18}$$

Therefore, for $\rho_a, \rho_b$ two irreps of $G$, $\rho_c$ is in the TD of $\rho_a \otimes \rho_b$ if and only if $\langle \chi_{\rho_a \otimes \rho_b}, \chi_{\rho_c} \rangle_G \neq 0$. Let us apply this to the 2D irreps of $D_n$ ($n > 2$). Let $\rho_i, \rho_j, \rho_k$ be three irreps defined as in (13). Notice that we have $\mathcal{X}_\rho(a^l x) = 0$. This yields

$$\langle \mathcal{X}_{\rho_i \otimes \rho_j}, \mathcal{X}_{\rho_k} \rangle = \frac{1}{2n} \sum_{l=1}^n \mathcal{X}_{\rho_i \otimes \rho_j}(a^l)\mathcal{X}_{\rho_k}(a^l)$$

$$= \frac{1}{2n} \sum_{l=1}^n \mathcal{X}_{\rho_i}(a^l)\mathcal{X}_{\rho_j}(a^l)\mathcal{X}_{\rho_k}(a^l)$$

$$= \frac{1}{2n} \sum_{l=1}^n 8\cos(\omega_l i)\cos(\omega_l j)\cos(\omega_l k)$$

$$= \frac{1}{n} \sum_{l=1}^n \cos(\omega_l(i+j+k)) + \cos(\omega_l(i+j-k))$$

$$+ \cos(\omega_l(j-i+k)) + \cos(\omega_l(j-i-k)).$$

Recall that $\sum_{l=1}^n \cos(\omega_l m) = \begin{cases} 0 \text{ if } m \neq 0 \\ n \text{ if } m = 0 \end{cases}$. Therefore, if we assume $0 \leq i \leq j$ without loss of generality, $\langle \mathcal{X}_{\rho_i \otimes \rho_j}, \mathcal{X}_{\rho_k} \rangle \neq 0$ if and only if

$$\begin{cases} k = i + j \text{ or} \\ k = j - i. \end{cases}$$

Therefore, based on Definition 2.3, by utilizing $\beta_{\rho_i,\rho_j}$, $\mathcal{F}(\Theta)_{\rho_i}$ and $\mathcal{F}(\Theta)_{\rho_j}$, we can compute $\mathcal{F}(\Theta)_{\rho_k}$ for $k \in \{i+j, \ j-i\}$. By iterating, if $\mathcal{F}(\Theta)_{\rho_1}$ is known, $\beta_{\rho_1,\rho_1}$ gives $\mathcal{F}(\Theta)_{\rho_2}$. Then, $\beta_{\rho_1,\rho_2}$ can be leveraged to obtain $\mathcal{F}(\Theta)_{\rho_3}$. Continuing the procedure provides $\mathcal{F}(\Theta)_{\rho_k}$ for $k = 2, ..., \lfloor \frac{n-1}{2} \rfloor$ by using $\beta_{\rho_1,\rho_{k-1}}$. We have thus obtained the Fourier coefficients associated to all the 2D irreps.

It remains to show that the iteration also recovered the Fourier coefficients associated to the 1D irreps. This is because $\rho_{01}$ is in the TD of $\rho_1 \otimes \rho_1$ and, for $n$ even, $\rho_{02}, \rho_{03}$ are in the TD of $\rho_1 \otimes \rho_{\frac{n}{2}-1}$. Indeed,

$$\langle \mathcal{X}_{\rho_1 \otimes \rho_1}, \mathcal{X}_{\rho_{01}} \rangle = \frac{1}{2n} \sum_{l=1}^n \mathcal{X}_{\rho_1 \otimes \rho_1}(a^l)\mathcal{X}_{\rho_{01}}(a^l)$$

$$= \frac{1}{2n} \sum_{l=1}^n 4\cos(\omega_l)^2$$

$$= \frac{1}{n} \sum_{l=1}^n 1 + \cos(2\omega_l) = 1.$$

Moreover, for $n$ even and $\rho \in \{\rho_{02}, \rho_{03}\}$, we have

$$\langle \mathcal{X}_{\rho_1 \otimes \rho_{\frac{n}{2}-1}}, \mathcal{X}_\rho \rangle = \frac{1}{2n} \sum_{l=1}^n \mathcal{X}_{\rho_1 \otimes \rho_{\frac{n}{2}-1}}(a^l) \mathcal{X}_\rho(a^l)$$

$$= \frac{1}{n} \sum_{l=1}^n 1 + \cos\left(\omega_l \left(\frac{n}{2} - 2\right)\right) (-1)^l$$

$$= 1.$$

In conclusion, the procedure of Algorithm 4 recovers all the Fourier coefficients and the selective $G$-Bispectrum.

### E.2 The Clebsch-Gordan matrices on $D_n$

The matrix algebra properties that we use in this subsection can be found, e.g., in [1]. Recall from Theorem A.12 the (implicit) definition of the Clebsch-Gordan matrices:

$$(\rho_1 \otimes \rho_2)(g) = C_{\rho_1, \rho_2} \left[ \bigoplus_{\rho \in \mathcal{R}} \rho(g) \right] C_{\rho_1, \rho_2}^\dagger, \tag{19}$$

where $C_{\rho_1, \rho_2}^\dagger C_{\rho_1, \rho_2} = I$. We only consider the case of $\rho_1, \rho_2$ both 2d irreps of $D_n$ since otherwise, the Clebsch-Gordan matrix is the scalar 1. First notice that $(\rho_1 \otimes \rho_2)(g)$, is an orthonormal matrix. Indeed, using the properties of the Kronecker product, we obtain ("$(g)$" omitted for clarity):

$$(\rho_1 \otimes \rho_2)(\rho_1 \otimes \rho_2)^\dagger = (\rho_1 \otimes \rho_2)(\rho_1^\dagger \otimes \rho_2^\dagger)$$

$$= (\rho_1 \rho_1^\dagger) \otimes (\rho_2 \rho_2^\dagger)$$

$$= I \otimes I = I$$

For $Q$ a real orthogonal matrix ($Q^T Q = I$), and $V S V^T$ with $V^T V = I$, a real Schur decomposition of $Q$, it is known that $S$ is block diagonal with blocks of size $1 \times 1$ or $2 \times 2$. These blocks are themselves orthogonal matrices. Therefore, the real Schur decomposition is the decomposition in (19) up to permutations. In order for $S$ to represent exactly the irreps from (13), the non-zero sub-diagonal elements should all be positive. If not, the symmetric element is positive and a permutation $P$ must be added to exchange their positions: $Q = (VP)(P^T S P)(VP)^T$. $P$ permutes the two columns of $V$ associated with the permuted $2 \times 2$ block of $S$.

---

**Algorithm 5** Compute Clebsch-Gordan matrices on $D_n$

---

1: **Input:** $\rho_1, \rho_2$, two 2d irreps of $D_n$.
2: Pick any $g \in D_n$, e.g., $g = a$.
3: Compute $(\rho_1 \otimes \rho_2)(g) = (VP)(P^T S P)(VP)^T$, a valid real Schur form.
4: Set $C_{\rho_1, \rho_2} = VP$.
5: Set $\bigoplus_{\rho \in \mathcal{R}} \rho(g) = P^T S P$
6: **Return:** $C_{\rho_1, \rho_2}, \bigoplus_{\rho \in \mathcal{R}} \rho(g)$.

---

## F  Bispectrum inversion for octahedral and full octahedral groups

We provide a sketch of the procedure to retrieve $\mathcal{F}(\Theta)$ given $\beta(\Theta)$ for the octahedral group and the full octahedral group. These two groups are available in the `escnn` library. These groups are easier to deal with than the cyclic and dihedral groups presented in the paper, given that they do not come from a *family* of groups. Indeed, our proofs for the cyclic (resp. dihedral) groups needed to work for *all* cyclic groups $C_N$, and for all dihedral groups $D_N$, for all $N$. The octahedral and full octahedral groups are only two groups.

| $\otimes$ | $\rho_0$ | $\rho_1$ | $\rho_2$ | $\rho_3$ | $\rho_4$ |
|---|---|---|---|---|---|
| $\rho_0$ | 10000 | 01000 | 00100 | 00010 | 00001 |
| $\rho_1$ | 01000 | 11110 | 01111 | 01100 | 00100 |
| $\rho_2$ | 00100 | 01111 | 11110 | 01100 | 01000 |
| $\rho_3$ | 00010 | 01100 | 01100 | 10011 | 00010 |
| $\rho_4$ | 00001 | 00100 | 01000 | 00010 | 10000 |

Table 4: Kronecker table of the octahedral group using `escnn`. For the binary word at position $i, j$ in the table, the $k$th 'letter' is 1 if $\rho_k \in \rho_i \otimes \rho_j$, 0 otherwise.

### F.1 Octahedral group

The octahedral group has 24 elements and 5 irreps. We can compute its Kronecker table, either manually using characters $\chi$ or using a Python package such as `escnn`. We give its Kronecker table below (Table 4), where each column/row represents one irrep, labelled $\rho_0, \rho_1, ..., \rho_4$.

We apply the procedure from Algorithms 2, 3 and 4 to Table 4. This procedure relies on the use of Theorem 2.3. We first select the bispectral coefficient $\beta_{\rho_0,\rho_0}$ (we omit "($\Theta$)" for clarity) to get the component $\mathcal{F}_{\rho_0}$ where $\rho_0$ is the trivial representation. Next, we choose $\beta_{\rho_0,\rho_1}$ and use $\mathcal{F}_{\rho_0}$ to obtain $\mathcal{F}_{\rho_1}$ (we know from Table 4 that $\rho_1 \in \rho_0 \otimes \rho_1$) up to an indeterminacy which is a transformation in $O(3)$ and corresponds to the indeterminacy factor from Appendix B. Then, we select $\beta_{\rho_1,\rho_1}$ to get the Fourier components $\mathcal{F}_{\rho_2}, \mathcal{F}_{\rho_3}$. Lastly, we select $\beta_{\rho_1,\rho_2}$ to get the missing Fourier component $\mathcal{F}_{\rho_4}$.

In summary, we only need 4 bispectral coefficients ($\beta_{\rho_0,\rho_0}, \beta_{\rho_1,\rho_0}, \beta_{\rho_1,\rho_1}, \beta_{\rho_1,\rho_2}$) instead of $5^2 = 25$ in order to get the five Fourier components, i.e., the full Fourier transform of the signal. In total, this involves $1 + 9 + 81 + 81 = 172$ scalar coefficients.

### F.2 Full octahedral group

| $\otimes$ | $\rho_0$ | $\rho_1$ | $\rho_2$ | $\rho_3$ | $\rho_4$ | $\rho_5$ | $\rho_6$ | $\rho_7$ | $\rho_8$ | $\rho_9$ |
|---|---|---|---|---|---|---|---|---|---|---|
| $\rho_0$ | 1000000000 | 0100000000 | 0010000000 | 0001000000 | 0000100000 | 0000010000 | 0000001000 | 0000000100 | 0000000010 | 0000000001 |
| $\rho_1$ | 0100000000 | 1111000000 | 0111100000 | 0110000000 | 0010000000 | 0000001000 | 0000011110 | 0000001111 | 0000001100 | 0000000100 |
| $\rho_2$ | 0010000000 | 0111100000 | 1111000000 | 0110000000 | 0100000000 | 0000000100 | 0000011111 | 0000011110 | 0000001100 | 0000001000 |
| $\rho_3$ | 0001000000 | 0110000000 | 0110000000 | 1001100000 | 0001000000 | 0000000010 | 0000001100 | 0000001100 | 0000010011 | 0000000010 |
| $\rho_4$ | 0000100000 | 0010000000 | 0100000000 | 0001000000 | 1000000000 | 0000000001 | 0000000100 | 0000001000 | 0000000010 | 0000010000 |
| $\rho_5$ | 0000010000 | 0000001000 | 0000000100 | 0000000010 | 0000000001 | 1000000000 | 0100000000 | 0010000000 | 0001000000 | 0000100000 |
| $\rho_6$ | 0000001000 | 0000011110 | 0000011111 | 0000001100 | 0000000100 | 0100000000 | 1111000000 | 0111100000 | 0110000000 | 0010000000 |
| $\rho_7$ | 0000000100 | 0000001111 | 0000011110 | 0000001100 | 0000001000 | 0010000000 | 0111100000 | 1111000000 | 0110000000 | 0100000000 |
| $\rho_8$ | 0000000010 | 0000001100 | 0000001100 | 0000010011 | 0000000010 | 0001000000 | 0110000000 | 0110000000 | 1001100000 | 0001000000 |
| $\rho_9$ | 0000000001 | 0000000100 | 0000001000 | 0000000010 | 0000010000 | 0000100000 | 0010000000 | 0100000000 | 0001000000 | 1000000000 |

Table 5: Kronecker table of full octahedral group using `escnn`. For the binary word at position $i, j$ in the table, the $k$th 'letter' is 1 if $\rho_k \in \rho_i \otimes \rho_j$, 0 otherwise.

The full octahedral group has 48 elements and 10 irreps. Again, we can compute its Kronecker table using a Python package such as `escnn`. We give its Kronecker table below (Table 5), where each column/row represents one irrep, labelled $\rho_0, \rho_1, ..., \rho_9$.

We apply the procedure from Algorithms 2, 3 and 4 to Table 5. Again, this procedure relies on the use of Theorem 2.3. $\beta_{\rho_0,\rho_0}$ (we omit "($\Theta$)" for clarity) allows to compute $\mathcal{F}_{\rho_0}$ directly, such as in Algorithm 4. Then, from $\beta_{\rho_0,\rho_6}$, we obtain $\mathcal{F}_{\rho_6}$ up to an unknown group action in $O(3)$. Then, using $\beta_{\rho_6,\rho_6}$ and $\mathcal{F}_{\rho_6}$, we obtain $\mathcal{F}_{\rho_1}, \mathcal{F}_{\rho_2}$ and $\mathcal{F}_{\rho_3}$. Next, leveraging $\beta_{\rho_1,\rho_2}, \mathcal{F}_{\rho_1}$ and $\mathcal{F}_{\rho_2}$, we obtain $\mathcal{F}_{\rho_4}$. Using $\beta_{\rho_1,\rho_6}, \mathcal{F}_{\rho_1}$ and $\mathcal{F}_{\rho_6}$, we obtain $\mathcal{F}_{\rho_5}, \mathcal{F}_{\rho_7}, \mathcal{F}_{\rho_8}$. Finally, with $\beta_{\rho_1,\rho_7}, \mathcal{F}_{\rho_1}$ and $\mathcal{F}_{\rho_7}$, we obtain the last coefficient $\mathcal{F}_{\rho_9}$. Hence we have recovered all the Fourier coefficients using only $\beta_{\rho_0,\rho_0}$, $\beta_{\rho_0,\rho_6}, \beta_{\rho_6,\rho_6}, \beta_{\rho_1,\rho_2}, \beta_{\rho_1,\rho_6}, \beta_{\rho_1,\rho_7}$, thus a total of 6 bispectral coefficients instead of 100. In terms of scalar values, this involves $1 + 9 + 81 + 81 + 81 + 81 = 334$ coefficients.

## G Training of the $G$-CNN architecture

The SO(2)-MNIST/EMNIST datasets are obtained after applying random planar rotations on each image of the datasets MNIST [23], EMNIST [5] respectively. In the case of O(2)-MNIST/EMNIST, in addition to a planar rotation, a reflection is applied with probability $\frac{1}{2}$. The original size of each image is conserved. The size of the training sets are 60 000 and 88 800 for MNIST and EMNIST, respectively.

We conserve the architecture of [28]. For all invariant layers, being the $G$-TC, the selective $G$-Bispectrum and the Avg/Max $G$-pooling, the architecture is composed of a $C_8/D_8$-convolutional block with $K$ filters (see Table 2). Then, the invariant layer is applied before feeding the output to a MLP. The MLP is composed of 3 fully-connected layers with ReLU non-linearity. A final fully-connected linear layer is applied for classification. The vector of the output sizes of these layers is given by $[o1, o2, o3, ol]$ respectively. $o2 = o3 = 64$ and $ol$ is equal to the number of classes of the dataset. $o1$ is tuned to reach the parameter count from Table 2.

