# OpenReview forum: "The Selective $G$-Bispectrum and its Inversion: Applications to $G$-Invariant Networks"
_NeurIPS.cc/2024/Conference — NeurIPS 2024 poster_

### Official Review · Reviewer_Wppf · 2024-07-15

**Soundness:** 2
**Presentation:** 3
**Contribution:** 3
**Rating:** 4
**Confidence:** 5

**Summary:**

The paper proposes a novel invariant layer for group-convolution networks (GCNNs) based on the "selective bi-spectrum".
With respect to the expensive bispectrum which has $O(|G|^2)$ coefficients, the selective bispectrum only contains a subset of $O(|G|)$ coefficients which still ensure the completeness of the invariant features and can be computed in $O(|G| log |G|)$ when a FFT algorithm is available.
Some preliminary experiments show the proposed method has comparable runtime and performance to the typically used max-pooling operator.

**Strengths:**

The manuscript is clearly written and the proposed method is original and interesting.
The paper also includes some novel theoretical results, which could be useful for a wider audience.
The empirical validation is in its very early stage, but the preliminary results are encouraging.

**Weaknesses:**

I feel like the empirical evaluation of the method is still too weak for acceptance.

First, the authors only experiment with variations of the MNIST dataset: while this is a good dataset for some preliminary experiments to validate an idea is a simple and easily reproducible environment, I don't think it is sufficient to draw significant conclusions, especially about properties such as expressiveness of an operator or completeness of a representation.
The authors should consider experimenting with more realistic datasets: for example, previous works typically used medical datasets, which still show rotational symmetries but provide higher resolution images and more challenging classification tasks.
Because the contributions of this paper include the theoretical description of the selective bispectrum for some 3D groups, the authors should include some experiments on 3D datasets to validate that too; these datasets could also provide more convincing evidence of the value of the proposed method.

Second, even if larger datasets were used, the current experimental setting doesn't provide conclusive answers on when one should use max-pooling or bi-spectrum.
Indeed, the comparison in Fig. 5 is a bit misleading as the accuracy is plotted as a function of the number of filters K, but the computational cost of the two methods is different when varying K.
Instead, a plot showing the accuracy vs (some proxy for) the computational cost seems a more suitable choice, drawing a more complete picture and quantifying the trade-offs discussed between lines 259-263.


Overall, while I am very positive about the proposed idea, I think that deserves a much more thorough empirical validation.

**Questions:**

Fig. 6: this seems just a qualitative results useful to demonstrate the robustness of the selective bispectrum, but a proper quantitative validation of the robustness is still missing. A more comprehensive and thorough comparison between selective bispectrum, normal bispectrum and, maybe, max-pooling seems necessary. Why not reporting some statistics aggregated over a much larger dataset of random signals rather than visualising just 6 examples? It would be even more convincing to do so using the deep features extracted by a GCNN from a dataset of natural images, to prove the completeness is relevant beyond simple synthetic settings.


Theorem 4.5: Is it correct that the (4) G-Bispectral coefficients are matrices? I find the choice of counting the matrices instead of individual scalars to be a bit misleading, as these matrices have very different shapes, making this number not very informative. Why not reporting only the number of scalar coefficients? This is a quantity which is meaningful across different groups and that can be directly compared with the group size |G| (the Fourier transform over the group G always has |G| coefficients, regardless of which finite group we consider).


Theorem 2.4: alpha is used but never defined

Table 1: why is K not included inside the big-O notation?


Line 242: "... so that the G-Bispectrum scales worth". This sentence is a bit unclear.

**Limitations:**

The authors could discuss practical challenges in an efficient or parallel implementation of the selective bispectrum operator due to the sparsity and irregular shapes of the matrices involved.

---

> ### Author Rebuttal · Authors · 2024-08-07
>
> Dear Reviewer Wppf,
>
> We thank you very much for your review and for your time. We address your comments and questions below.
>
> ### Experiments on larger datasets
>
> We address this concern in our global response to all reviewers.
>
> ### Accuracy versus computational cost
>
> We agree that showing accuracy versus computational cost is necessary to make our main point much stronger. We have added a scatter plot quantifying this trade-off for the different pooling methods. We describe it in our global reponse to all reviewers. We thank you for the great suggestion.
>
> ### Question on evaluation of robustness/completeness
>
> Thank you for raising this important point. We appreciate your question, as it helped us realize that the purpose of Figure 6 was not clearly presented. We clarify it here, and also refer you to the additional experiments on robustness/completeness that we discuss in our global response to all reviewers.
>
> Figure 6 is _not_ meant to be a _validation_ of the completeness of the G-Bispectrum, but rather an _illustration_. This is why, as you correctly noted, Figure 6 is qualitative and not quantitative.
>
> Our point here is that Theorem 4.1 (completeness of the selective G-bispectrum on $C_n$) is known to be true since it is proved: all signals that have the same selective $G$-bispectrum are known to be identical up to group action. Hence, numerical experiments can only validate it. In consequence, Fig. 6 is not a statistical validation but an illustrative example to convince the reader. This is why we visualize examples and do not report statistics aggregated over a much larger dataset of random signals.
>
> Besides, to provide a more comprehensive picture of the robustness of the selective $G$-bispectrum, we have added in the "Answer to all reviewer" and the PDF attached an experiment where, this time, input image are recovered.
>
> We hope this re-framing of Figure 6 clarifies its purpose. We will integrate this wording into our revised manuscript.
>
> Additionally, we provide novel experiments illustrating robustness and completeness in our global response to all reviewers.
>
> ### Question on memory output: report the number of scalar coefficients
>
> Yes, the G-bispectral (Bsp) coefficients are matrices in general, although they are scalars for commutative groups. We originally chose to report the number of G-bispectral coefficients to report a single value across groups, and also because the size of these matrices does not change the O complexity. Yet, you are absolutely right that it would be very useful (and potentially less misleading) to report the number of scalar coefficients as well. We do so below.
>
> The number of scalar coefficients in the G-TC is $K * |G| * \frac{|G| + 1}{2}$. This formula applies to all the groups discussed on our paper: $C_n$, $D_n$ the octahedral group $O_h$ and the full octahdral group $FO_h$.
>
> The number of scalar coefficients in the Max G-pool is $K$, also the same across the groups $C_n$, $D_n$, $O_h$ and $FO_h$.
>
> The number of scalar coefficients in the selective G-Bsp varies depending on the group.
>   - $C_n$: $K * |G|$ scalar coefficients,
>   - $D_n$: $\approx K * 4|G|$ scalar coefficients,
>   - $O_h$: $K * 172$ scalar coefficients ($\approx K* 7 |G|$),
>   - $FO_h$: $K*334$ scalar coefficients ($\approx K *7 |G|$).
>
> These numbers are computed by considering which the G-Bsp coefficients are used in the selective G-Bsp (which is given in our theorems and proofs), and by summing the scalar coefficients of their respective matrices.
>
> ### Discussion on practical challenges in parallel implementation
>
> Thank you for raising this point. Our main contribution is about computational gains, made possible thanks to new mathematical theorems on group theory. Thus, we agree that it is important to discuss other ways to obtain computational gains. Parallel computations is one of them. Following your suggestion, we discuss this point below. (Notation: Bsp = Bispectrum)
>
> To further gain in computational efficiency, we could compute the G-Bsp coefficients in parallel. This can be done both for the full G-Bsp and for the selective G-Bsp.
>
> Indeed, for both, we know _which_ G-Bsp coefficients are needed: we need all of them for the full G-Bsp (or, half of them, given the symmetry rho_i, rho_j <> rho_j, rho_i), and a subset of them for the selective G-Bsp where the subset is given by our theorems. Consequently, each G-Bsp coefficient can be computed with a separate process. One would need to account for the fact that each G-Bsp coefficient has a different memory footprint. Indeed, each G-Bsp coefficient is a matrix with a size depending on the nature of the coefficient.
>
> Since this parallelization would speed up both the full G-Bsp and the selective G-Bsp, the selective G-Bsp would still be faster. Lastly, since the matrices defining the G-Bsp coefficients are generally non-sparse, no further computational gain that can be obtained there.
>
> ### Notations
>
> Thank you for reporting the typos. We have corrected these.
>
> ### Conclusion
>
> Again, we thank you for the thorough review and the very interesting questions. Your feedback has been beneficial for us, and we believe that the additional experiments and discussions significantly strenghten our paper. If you have any remaining questions or comments, we would be happy to discuss further!
>
> We hope that we have addressed your concerns and we look forward to discussing these topics further.

---

> > ### Comment · Reviewer_Wppf · 2024-08-09
> >
> > I thank the authors for the detailed answer.
> >
> > I understand the authors' argument about the simple and controlled experiments, but I do not see that as a good reason why not **also** including a larger scale experiment to validate the method.
> > Indeed, **1)** it is common to test equivariant networks on much larger and more complex datasets than MNIST by augmenting them with group transformations and still ensuring that we know exactly which groups structure the data follows  (see again the recommendations in my original review).
> > **2)** I still believe that MNIST is not sufficient to draw strong conclusions about properties such as expressiveness of an operator or completeness of a representation, due to its simplicity.
> >
> > Unfortunately, while I am very positive about the theoretical value of this work, I think a stronger empirical evaluation - validating the important claims about the proposed method - is necessary for the acceptance of the paper.

---

> > > ### Author Response · Authors · 2024-08-09
> > >
> > > Dear reviewer WWpf,
> > >
> > > Thank you for reading and engaging with our rebuttal. We appreciate your time.
> > >
> > > While we respectfully maintain that large-scale dataset experiments may not be strictly necessary for theory-heavy research contributions, we understand and acknowledge that additional empirical validation could be valuable and convincing to you and potentially a larger class of our future potential readers.
> > >
> > > Consequently, to enhance the impact of this work, we will explore the possibility of conducting experiments on a larger dataset and will provide an update as soon as possible.

---

> > > > ### Author Response · Authors · 2024-08-12
> > > >
> > > > Dear reviewer Wppf, please find the results on SO2-CIFAR-10 as asked in an additional comment to all reviewers.
> > > >
> > > > We thank you again for your review and we would be glad if this could convince you to increase your score.

---

### Official Review · Reviewer_ZVp5 · 2024-07-16

**Soundness:** 2
**Presentation:** 3
**Contribution:** 2
**Rating:** 5
**Confidence:** 4

**Summary:**

The paper addresses the problem of achieving invariance to nuisance factors in signal processing and deep learning, particularly those describable by group actions (e.g., rotations, translations). The authors propose the selective G-Bispectrum, a computationally efficient variant of the G-Bispectrum, which reduces complexity from $O(|G|^2)$ to $O(|G|)$ in space and $O(|G| \log |G|)$ in time when an FFT is available on the group $G$.

**Strengths:**

The paper presents a novel approach to reducing the computational complexity of the G-Bispectrum, a key operation in achieving G-invariance in signal processing and deep learning. This reduction is significant, lowering the complexity from \(O(|G|^2)\) to \(O(|G|)\) in space and from \(O(|G|^2)\) to \(O(|G|\log|G|)\) in time, assuming the availability of an FFT on \(G\). The selective G-Bispectrum is a creative combination of existing ideas in group theory and signal processing, applied in a new way to enhance deep learning architectures. This work stands out by addressing computational limitations that have hindered the widespread adoption of the G-Bispectrum.

The paper is well-structured and thorough in its theoretical contributions. The authors provide rigorous proofs of the mathematical properties of the selective G-Bispectrum.

The significance of this work lies in its potential to advance the field of geometric deep learning and signal processing on groups. By reducing the computational complexity of the G-Bispectrum, the authors make this tool more practical for real-world applications.

**Weaknesses:**

The paper primarily evaluates the proposed selective G-Bispectrum on the MNIST and EMNIST datasets. While these datasets are standard benchmarks in machine learning, they are relatively simple and may not fully stress-test the proposed method’s effectiveness in more complex and varied scenarios, especially larger group such as 3D group.

Although the paper claims that the selective G-Bispectrum has higher computational efficiency, it does not explicitly compare the computational costs and performance together in the experiments. This omission undermines the argument for computational efficiency, particularly given that the performance of the selective G-Bispectrum is not as strong as the G-TC method.

**Questions:**

**Question:** It is mentioned in the paper that the selective G-Bispectrum offers greater selectivity and robustness. Could you provide experimental evidence to validate these claims?
**Suggestion:** Include experiments and analyses specifically designed to demonstrate the greater selectivity and robustness of the selective G-Bispectrum. This could involve:
- **Selectivity:** Experiments that show how well the method distinguishes between different classes or handles variations within the same class.
- **Robustness:** Tests under various noise conditions, transformations, or adversarial attacks to demonstrate the method's stability and reliability.


**Question:** Why were only MNIST and EMNIST datasets used for the evaluation? Would the selective G-Bispectrum perform similarly on more complex datasets?
**Suggestion:** Expand the experimental section to include more diverse and challenging datasets, such as CIFAR-10, ModelNet10. This will provide a more comprehensive assessment of the method's effectiveness.

**Limitations:**

yes

---

> ### Author Rebuttal · Authors · 2024-08-07
>
> Dear Reviewer ZVp5,
>
> We thank you for your time and thoughtful review. We address your comments and suggestions below.
>
> ### Experiments on larger datasets
>
> Thank you for raising this point. We comment on it in our global response to all reviewers.
>
> ### Showing accuracy versus computational cost
>
> We agree that showing accuracy versus computational cost would make our points much stronger. We have added a scatter plot quantifying this trade-off for the different pooling methods. See global response to all reviewers.
>
> ### Showing "selectivity"
>
> Indeed, we mention that the G-Bispectrum offers greater selectivity in our sentence:
>  "the $G$-Bispectrum has been incorporated into deep neural network architectures as a computational primitive for $G$-invariance - akin to a pooling mechanism, but with greater selectivity and robustness."
>
> In this sentence, we use the word "selectivity" to refer to the completeness property of the G-Bispectrum. The G-Bispectrum of a signal is complete in the sense that it removes only the variation due to the action of a group G on the signal, while preserving all information about the signal's structure: it _selectively_ removes the group action without destroying the representation. We have provided empirical evidence to validate this claim in our results presented in Figure 6 and the new experiment from the PDF attached.
>
> In contrast, we believe that you understood the word "selectivity" as it is commonly used in machine learning, where it refers to "specificity", the true negative rate (TNR) of a classification task. We apologize for our poor wording that led to this confusion.
>
> To avoid misunderstanding, we propose to rephrase our original sentence as follows:
> "the $G$-Bispectrum has been incorporated into deep neural network architectures as a computational primitive for $G$-invariance - akin to a pooling mechanism, but with greater selectivity and robustness."
>
> Since we do not claim greater "specificity" in the classification sense, we have not provided results on this front. However, we can show the confusion matrices of our classification results in supplementary materials. We hope that we have addressed this concern. Let us know if we should provide additional details here.
>
> ### Showing robustness
>
> Thank you very much for this suggestion. We have added an experiment in our global answer to all reviewers accordingly.
>
> ### Conclusion
>
> Again, we thank you for your time and attention. We hope that we have addressed your concerns and we look forward to discussing these topics further.

---

> > ### Author Response · Authors · 2024-08-12
> > **Additional experiments addressing your concern**
> >
> > Dear Reviewer ZVp5,
> >
> > Following our post-rebuttal discussion with reviewer Wppf, we have conducted additional experiments using the more complex CIFAR-10 dataset. These results are detailed in our global response to all reviewers and address one of your initial concerns.
> >
> > These experiments also complement our other new experiments on robustness and our addition of a cost-accuracy analysis plot – both of which were incorporated in response to your insightful feedback. Addressing your comments has been very beneficial to us, as these additions have strengthened our paper. We thank you for engaging with our work!
> >
> > If you agree that we have adequately addressed several of your original concerns, may we kindly ask if you would consider raising your score to support the acceptance of our paper?

---

### Official Review · Reviewer_UtSC · 2024-07-17

**Soundness:** 3
**Presentation:** 3
**Contribution:** 3
**Rating:** 5
**Confidence:** 3

**Summary:**

The work focuses on the problem of achieving invariance to extraneous nuisance variables in signal processing and deep learning by utilizing group actions such as rotations and translations.

The focus is on the G-Bispectrum, which is a tool used to extract signal characteristics that are invariant to such actions. This tool provides advantages similar to pooling techniques, but with more selectivity and robustness.

The G-Bispectrum has been limited in its application because to its high computational cost, which is O(|G|2). However, this study presents a strategy to lower this complexity to O(|G|) by proposing selective G-Bispectrum.

The novel methodology preserves the mathematical characteristics of the G-Bispectrum and enhances the precision and robustness of neural networks, while simultaneously delivering substantial speed ups.

**Strengths:**

The research presents a clear motivation, which is to decrease the computational complexity of the G-Bispectrum from O(|G|2) to O(|G|) without compromising its effectiveness as a computational primitive for G-invariance. The emphasis on enhancing computing efficiency while maintaining performance is clearly expressed and pertinent.

The proposed method is technically valid. The assumptions employed are rational, and the research establishes a robust theoretical basis for the selected G-Bispectrum, effectively showcasing its mathematical qualities such as G-invariance, completeness, and uniqueness.

 The paper is very well-written and easy to follow. Figure 1 enhances the clarity of the presentation by providing a comprehensive overview of the motivation for the proposed strategy and successfully conveying the essential themes.

The research presents extensive empirical evidence showing that the selective G-Bispectrum achieves significant improvements in speed compared to the whole G-Bispectrum in real-world scenarios. The enhancement in efficiency renders the selected G-Bispectrum a feasible and efficient substitute for attaining G-invariance in deep learning models.

**Weaknesses:**

The proposed approach is currently limited to discrete groups. Several contemporary equivariant networks have been applied in domains that involve continuous groups like $SO(3)$ and $SE(3)$. However, the current technique does not readily adapt to these situations, and the paper does not investigate or experiment with such extensions in these domains. Could the authors comment on this limitation and discuss whether it could be a potential direction for future research?

Furthermore, the tests conducted in this research are restricted to small-scale datasets such as MNIST. There is the possibility of analyzing larger and more intricate datasets in order to more effectively showcase the advantages of the suggested approach. I strongly urge the authors to contemplate using their methodology on larger-scale image datasets in order to demonstrate its efficacy in a more thorough manner.

**Minor**

In line 183, novel theorems, Although the coefficients may not be precisely zero, they could be in close proximity to zero. When performing numerical computations, this can result in problems related to stability and precision.

The formula presented in Theorem 4.4, which states that the number of bispectral matrix coefficients is equal to ⌊n-1/2⌋ + 2, may benefit from a more comprehensive explanation for readers who are less acquainted with dihedral groups.

The sentence in Theorem 4.4, "This corresponds to 1 + 4 + 16 · ⌊n-1/2⌋ ≈ 4|Dn| scalar values," may be clarified by providing a concise explanation of the derivation of this approximation.

**Questions:**

The proposed approach is currently limited to discrete groupings. Various contemporary equivariant networks have been utilized for continuous groups like $SO(3)$ and $SE(3)$. Could you provide an analysis of the constraints associated with applying your approach to these continuous categories? Do you anticipate any theoretical or practical obstacles, and do you view this as a promising avenue for future investigation?

Recommendation: It is advisable to evaluate the suggested technique on image datasets of greater magnitude in order to demonstrate its efficacy. To better showcase the benefits of your technique, it is advisable to conduct trials using established benchmarks or more intricate datasets.

Question: What is the difference between the suggested selective G-Bispectrum and contemporary equivariant networks tailored for continuous groups? Are there any particular benefits or drawbacks that might be emphasized in the context of ongoing group actions?

---

> ### Author Rebuttal · Authors · 2024-08-07
>
> Dear Reviewer UtSC,
>
> We thank you for your review and insightful comments and suggestions. We address your points below.
>
> ## Continuous Groups
>
> We thank you for asking the very interesting question of how to generalize to continuous groups.
>
> The formulas defining the full and selective G-Bispectra (G-Bsp) only depend on the G-Fourier transform. Since the G-Fourier transform can be defined for continuous groups, so can the G-Bsp. However, practical difficulties arise upon implementation.
>
> Indeed, the full G-Bsp has one G-bispectral coefficient per irreps of the group G. While discrete groups have a discrete number of irreps, continuous groups can have an infinite number of them. For example, SO(3) has an infinite number of irreps. Accordingly, because of memory constraints, we cannot compute the _full_ G-Bsp for SO(3), nor for other continuous groups with an infinite number of irreps.
>
> Interestingly, the _selective_ G-Bsp that we propose does not require all of the G-bispectral coefficients. Thus, it does not require to use all the irreps of the group G. As a result, it is possible that we could compute the _selective_ G-Bsp for continuous groups.
>
> Let us, therefore, assume that the selective G-Bsponly requires a finite number of irreps for a given continuous group G. In this case, we face another obstacle. This obstacle appears when we want to use this G-Bsp as the pooling mechanism of a G-convolutional neural net - as done in our classification experiments. Indeed, the G-Convolution that preceeds the (selective) G-Bsp is a so-called regular G-convolution (Group Equivariant Convolutional Networks. Cohen & Welling, 2016), which uses a discretization of the group G. Thus, the output $\Theta$ of the G-convolution would be a signal over a discrete/discretized group. Consequently, we would not be able to use the continuous selective G-Bsp there.
>
> To overcome this obstacle, one could think of using a steerable G-Convolution, instead of a regular one, since steerable G-convolutions apply to continuous groups (Steerable CNNs. Cohen & Welling, 2017). In our paper, we chose to use regular G-Convolutions because: (i) we are in fact interested in discrete groups and, (ii) a regular G-convolution outputs a signal $\Theta$ defined over a group - much needed to apply the definition of the G-Bispectrum - whereas a steerable G-Convolution outputs a more complicated mathematical object for which the G-Bispectrum needs to be generalized. We discuss this aspect below.
>
> Consider a group $G$ that is the semi-direct product $G=\mathbb{Z}_2 \ltimes H$ of the group of translations $\mathbb{Z}_2$ and a group $H$ of transformations that fixes the origin $0 \in \mathbb{Z}_2$. Consider the output of the steerable G-Convolution, which is a map $\Theta: \mathbb{Z}_2 \rightarrow \mathbb{R}^K$, where $K$ is the number of filters in the convolution. The signal $\Theta$ has the following mathematical property: $\Theta$ is a $K$-dimensional field over the grid $\mathbb{Z}_2$ that transforms according to a specific representation $\pi$ of the group $G$. This representation is induced by a representation $\rho$ of its subgroup $H$ on $R^K$ (Steerable CNNs. Cohen \& Welling, 2017), such that it is common to write $\pi=\mathrm{Ind}\_H^G \rho$. Consequently, the interaction between the signal $\Theta$ and the group $G$ is very different for the steerable case, compared to the regular case.
>
> - For a signal $\Theta$, the action of an element $(t, r) \in G = \mathbb{Z}^2 \ltimes H$ is:
> $[\pi\_{t, r} \Theta] (x) = \rho(r)\Theta((t, r)^{-1} x) \in \mathbb{R}^K.$
>
> - By contrast, when we use regular G-convolution, the action of an element $g \in G$ on each coordinate of the signal $\Theta$ is:
> $[g \ast \Theta] (g') = \Theta((g^{-1} g) \in \mathbb{R}.$
>
> The interaction between the signal $\Theta$ and the group $G$ is central to the definition of a G-Bsp. In fact, there is currently no definition of a G-bispectrum for the general steerable case. However, there is a definition of G-Bsp for signals defined over the homogeneous space of a group $G$ by Kakarala, which corresponds to a special steerable case for which $\rho$ is the trivial representation: $\rho(r) = r$ for all $r \in H$.
>
> Therefore, generalizing to continuous groups involves i) determining whether the selective G-Bsp only requires a finite number of coefficients, and for the classification experiments: ii) using a steerable G-convolution with trivial $\rho$ together with the G-Bsp for homogeneous spaces.
>
> These ideas could provide fruitful paths for exploration in future work. We believe, nonetheless, that presenting the theory for discrete groups is an important first step and should remain the focus on our work. Relatedly, in geometric deep learning, the paper Group-Equivariant Convolution Networks (Cohen & Welling, 2016) introduced the G-CNN for discrete groups. Only later did the paper Steerable CNNs (Cohen & Welling 2017) present a theory that applies to continuous groups.
>
> We will gladly add this discussion to the final version of the paper, thank you for this suggestion.
>
> ## Larger datasets
>
> Thank you for raising this point. We discuss it in the global answer to all reviewers.
>
> # Minor
> - You are right to mention that numerical zeros are a recurrent issue in the field of numerical computations, especially when used for division. However, the G-Bsp relies on additions and multiplications. Hence, we did not encounter such problems in our experiments.
> - Thank you for raising this point about the number of scalar coefficients. In the revised paper, we add a paragraph clarifying the number of bispectral matrices and the related number of scalar coefficients: see our answer to reviewer Wppf.
>
> Specifically, the approximation you mention comes from the fact that $|D_n |=2n$. Hence, $16\frac{n-1}{2} \approx 4|D_n|$.
> # Conclusion
>
> Again, we thank you for your time and attention. We hope that we have addressed your concerns. We look forward to further discussing these topics.

---

> > ### Author Response · Authors · 2024-08-12
> > **Correcting a typo + Additional experiments**
> >
> > Dear Reviewer UtSC,
> >
> > We would like to correct a typo in our discussion explaining how we can extend our approach to continuous groups, answering a very interesting question from your review. The trivial representation rho is rho(r) = 1 and not rho(r) = r, such that the corrected sentence is: “which corresponds to a special steerable case for which ρ is the trivial representation:  ρ(r)=1 for all r∈H”.
> >
> > We apologize for this typo. We hope that our discussion of the extension to continuous groups picked your interest.
> >
> > We would also like to point you to our global answer to all reviewers which contains additional experiments on larger datasets (CIFAR-10).
> >
> > We thank you for your constructive feedback, which has been very beneficial to us as it has stimulated very interesting discussions.
> >
> > If you believe that we have adequately addressed your concerns, would you consider increasing your score to support the acceptance of our work?

---

### Official Review · Reviewer_P7yz · 2024-07-22

**Soundness:** 3
**Presentation:** 2
**Contribution:** 3
**Rating:** 7
**Confidence:** 3

**Summary:**

The paper proposes a new invariant layer for group convolutional neural networks (G-CNN). The proposed layer computes the spectral coefficients of the input data, that is based on higher-order spectral analysis: the bispectrum, which is the Fourier transform of the triple correlation (G-TC). While there can be up to |Irrep|^2 different coefficients in the bispectrum, the main contribution of the work is an algorithm that finds a subset of at most |G| of these coefficients. For specific groups (general abelian, dihedral, (full) octahedral), the authors proved that there exists a subset of size at most |G| that can reconstruct the input data (signal) perfectly, and that their algorithm, with the right choice of \rho_1, can find this set. Experiments are conducted on MNIST and EMNIST, comparing the newly proposed layer with max/sum pooling and G-TC layer.

**Strengths:**

- Novelty. The use of higher-order spectral analysis is relatively new in designing deep learning architecture over group and can be powerful since they are originally used to capture nonlinear/autocorrelational structure of the input signal. Capturing nonlinearity is nontrivial in designing G-CNNs as pointwise nonlinearity is known to not be the most appropriate choice of nonlinearity.
- Theoretical contributions, especially to nonabelian groups such as Dn and O are interesting and has substantial savings in the number of necessary coefficients to be efficient in practice. I did not go through the proofs line by line but believe that the proof idea is sound.
- Self-contained. The paper, together with the appendix is relatively self-contained in introducing group-theoretic and representation-theoretic concepts. The overall exposition/English is also easy to follow.
- Clear experiments and diagrams. The paper demonstrate a regime of small number of filters where their methods work much better than simple group pooling, which is a clear and instructive takeaway. Diagrams to demonstrate algorithms are also clear and make for a quicker read.

**Weaknesses:**

The main issues I have with the current manuscript is its motivation and some ill-defined notion.

Issues with ill-defined notions:
- The term "selective G-bispectrum" is not well-defined, despite being the central notion of the whole paper. The authors defined this notion by giving an algorithm that depends crucially on a hyperparameter \rho_1. The authors themselves have noted that the choice of \rho_1 will make or break the resulting set of coefficient (being a complete system). Thus, completeness is a property of the algorithm + choice of \rho_1 and it does not make sense to claim, for e.g., that the selective G-bispectrum is complete without specifying this choice of irrep.
- Algorithm 1 depends crucially on computing the Clebsch–Gordan matrices, which may be very costly in general for nonabelian groups. The authors have also acknowledged this point.

Issues with motivations:
- The authors motivate the paper by arguing that a complete (Definition in Theorem 2.4) invariance layer is more desirable than simple G-pooling layer (such as max or sum), which is an intuitive point. However, in the experiments, average-pooling, which is arguably more complete than max-pooling (or, to quote, less "excessively invariant") performs consistently worse than anything else, and to a large extent. Is there a good explanation for this?
- The proofs of all completeness theorems essentially boils down to being able to reconstruct the group FT coefficients of the input signal from the subset of bispectrum coefficient, which begs the questions of what happens if you simply input these |Irrep| numbers of group FT coefficients as an invariant layer, instead of going to the bispectrum and hoping that it can reconstruct group FT coefficients? The answer to this probably lies in the reference "The work of Kakarala (1992) illuminated the relevance of the G-Bispectrum for invariant theory, as it is the lowest-degree spectral invariant that is complete (Sturmfels, 2008)" but scheming through Sturmfels' book did not lead me to any such conclusion. Can the authors provide a more specific reference, and/or discuss this point (why bispectrum but not just FT coefficients, in light of completeness) a bit more? This would be an interesting experiments to look at too.
- Lastly, why spectral? Usually, signal processing literature makes use of spectral domain because of sparsity in that domain. The authors have also shown that it is faster to compute G-bispectrum coefficients than G-TC. These motivations (and perhaps others, if presence) to using spectral domain should be written up more clearly in the beginning of the paper.

There are also various typos (e.g. line 26, R not defined in line 493, ...) that I did not carefully keep track of in the interest of time.

**Questions:**

See weakness section.

**Limitations:**

The authors have acknowledged some of the limitations in their methods and definitions (detailed in Weakness section). There are no potential negative societal impact that are specific to this work and are worth highlighting.

---

> ### Author Rebuttal · Authors · 2024-08-07
>
> Dear reviewer P7yz, we wish to thank you again for your time. In what follows, we address the points you made in your review.
>
> # Issues with ill-defined notions:
>
> ## The term "Selective G-Bispectrum"
>
> You are right to mention that the "Selective G-bispectrum" admits degrees of freedom, that is, it is not uniquely defined and there are several "Selective G-bispectra". As you correctly pointed out, each choice of $\rho_1$ will yield a different "Selective G-bispectrum".
>
> In the revised version of the paper, we propose to clarify this point as follows. For the most common groups appearing in our theorems, we call "Selective G-bispectrum" _the_ bispectrum proposed in their respective proofs. Indeed, each theorem provides a precise set of G-bispectral coefficients. The Selective G-bispectrum will refer to that set. For other groups, we call Selective G-bispectra the G-bispectra obtained via our algorithmic procedure that yield the maximum number of G-Fourier coefficients.
>
> We thank you for this important remark on the main term of our paper. We will make sure to clarify this in the final version of our work.
>
> ## Computing the Clebsch-Gordan matrices
>
> You are correct mentioning that obtaining the Clebsch-Gordan (CG) matrices is not an easy task. However, this work needs to be done only once before the training. This is because these matrices are not data-dependent and can be stored. In consequence, obtaining the CG matrices is not a bottleneck.
>
> Additionally, we point out that - for the octahedral and the full octahedral groups - we only use the CG matrices in the proofs of our theorems. Specifically, we only use them to find out the minimal set of G-bispectral coefficients that is needed. In practice, when we implement the selective G-bispectra for these two groups, we do not even use the CG matrices. This is because the G-bispectral coefficients can be written in the following alternative form:
>
> $$\beta(\Theta)\_{\rho_1,\rho_2} = \sum\_{g_1 \in G} \sum_\{g_2 \in G} \sum\_{g \in G} \Theta^*(g)  \Theta\left(g g_1\right) \Theta(g g_2)[ \rho_1(g_1)^{\dagger} \otimes \rho_2(g_2)^{\dagger}].$$
>
> This expression comes from the definition of the G-bispectrum as the G-Fourier transform of the G-triple correlation.
>
> Since computational gain is an important part of our contribution, we feel that it is important that we discuss and clarify this important point in the final version of the paper. We thank you for this feedback.
>
> # Issues with motivations:
>
> ## Completeness of avg pooling versus max pooling
>
> We respectfully think there might me a confusion with the notion of completeness. You are right to notice that average pooling uses all data to be computed, however so does the max pooling since all data must be compared to retrieve the maximum. In consequence, the average of a list is not more complete than the maximum, one does not give more information about the distribution of a list than the other.
>
> ## Distinction between the G-bispectrum and the G-Fourier transform
>
> We are glad to precise the interest of the G-bispectrum w.r.t the Fourier transform (FT). The advantage of the G-bispectrum is its invariance to group action while the FT is not invariant: the FT is only equivariant. To achieve group-invariance of a neural network, the FT is thus not a interesting quantity, at least it does not provide specific improvement towards this goal.
>
> Kakarala (1992) proved the invariance of the $G$-Triple Correlation (G-TC) for any compact group $G$, and by extension, the invariance of the $G$-bispectrum. This line of research also proves that the G-TC is the unique, lowest-degree polynomial invariant map that is also complete. Consequently, to achieve completeness, the simplest approach is the use of the G-TC or of its Fourier transform: the G-bispectrum. This comment raises the follow-up question: why using the G-Bispectrum over the G-TC? This question is linked to your question "why a spectral approach?", which we answer below.
>
> ## Why a spectral approach?
>
> This is a very interesting question. Our motivation for proposing, and using, the "Selective G-bispectrum" is motivated by the $O(n\log(n))$ complexity that it allows to obtain. By comparison, the complexity of the G-TC is $O(n^3)$ and we were not able to design a "selective G-TC" to reduce it. Consequently, the selective G-bispectrum is the unique, lowest degree polynomial invariant map that is also complete and that is the most computationally efficient. We will insist more on this point in the final version of the paper.
>
> # Conclusion
>
> Again, we thank you for the attention given to our work and for your thorough review. We have seeked to address your concern and comments with this rebuttal. If the elements provided above enhance your appreciation for our work, may we kindly ask whether you would consider defending our paper and increasing your score?

---

> > ### Comment · Reviewer_P7yz · 2024-08-14
> >
> > Thanks the author for the detailed reply and addressing my points.
> >
> > I personally think the paper is at acceptable quality for publication at the conference based on the theoretical contribution alone. I  agree with other reviewers that the experiments (though clear and succinct) are still at 'proof-of-concept' stage (which explains why I'm not raising the score to award level). To make meaningful experimental contributions beyond verifying the theory, a lot more work would need to be done, such as ablation and testing on various architectures as well as larger datasets and on more varied tasks. This would make a nice extension perhaps for a journal publication but at the current stage of the manuscript, I believe keeping the current score is reasonable.

---

> > > ### Author Response · Authors · 2024-08-14
> > >
> > > Dear Reviewer P7yz,
> > > Thank you very much for your kind words and continued support. We appreciate that you value our theoretical contributions and, overall, our work. We are grateful for the attention you have dedicated to the review process!

---

### Author Rebuttal · Authors · 2024-08-07

We thank the four reviewers for their time and attention. All reviewers found our manuscript very well-written and easy to follow. We find this very encouraging especially given its substantial theoretical components in group theory. The reviewers also appreciated the novelty and depth of our theoretical results, the rigor of our proofs, and their importance for signal processing as a whole. We thank all reviewers for recognizing this contribution.

We have provided responses to each reviewer for unique points made in the review. Here, we highlight a few points that appear across reviews, to avoid redundancy.

### Use of Simple, Controlled Datasets in our experiments

Reviewers ZVp5 and Wppf asked for the rationale behind using simple datasets for our experiments, and asked for experiments on more complex datasets.

We agree that the datasets we use here are relatively simple. We chose to use simple datasets in our analyses because they allow us to precisely control their mathematical structure. We generate our synthetic datasets by applying specific groups' actions to these data exemplars. This approach ensures that we know exactly which groups structure the data, enabling us to formulate theoretical expectations for the application of the selective G-bispectrum layer.

In contrast, the CIFAR image dataset (mentioned by the reviewers) contains many variations not attributable to group actions, such as lighting changes and general differences in appearance across exemplars within the same category. Therefore, we theoretically expect less benefit from using group-invariant and group-equivariant structures with this dataset. Our goal in this work was to empirically demonstrate the theoretically expected properties of our proposed layer, which we believe we have achieved using (i) strong theoretical contributions with several new theorems, and (ii) simple, controlled datasets.

### Robustness to adversarial attacks

Reviewers ZVp5 and Wppf asked for additional experiments to illustrate the robustness of the bispectrum. We first explain the link between completeness and robustness. Then, we provide new experiments and present their results here.

First, we emphasize that the completeness of the the triple correlation (TC), and its Fourier transform, the bispectrum,  is well-documented in signal processing literature (see Yellott & Iverson (1992) for translation-invariant case, and Kakarala (2009) for compact groups). The selective G-Bispectrum contains the same information as the G-bispectrum and is therefore also complete for the groups we consider. This completeness implies that it is inherently robust against invariance-based adversarial attacks. In such attacks, one finds non-identical inputs (up to group action) that produce identical output representations. Completeness prevents from this event, thereby ensuring robustness to these attacks.

We provide empirical illustration of this property with a new set of experiments. Specifically, we perform invariance-based adversarial attacks on the models trained in our experiments. Our results are presented in Figure 1 of the PDF attached. They show that the model with selective G-bispectrum is complete and robust to these attacks, whereas the Max G-Pool model is not. Specifically, when we optimize input images to match a target selective G-bispectrum, obtained form a target image, every optimized input image is identical to the target image up to group action. By contrast, an adversarial agent can find several input images that would yield a target output in the context of the Max G-pool model because it is not complete.

This analysis, initially established by Sanborn et al. (ICLR 2023), and later used in Sanborn & Miolane (2023) for the TC, empirically demonstrates the theoretically-expected completeness and robustness property in the trained model.

### Cost-Accuracy Trade-offs

Reviewers ZVp5 and Wppf raised the concern that the trade-off between computational cost and accuracy was not explicit enough, since computational cost and accuracy were on two separate figures in the manuscript. We agree and address this point here. We now plot computational cost and accuracy on the same figure to draw a complete, quantified picture of their trade-offs.

In the PDF attached, we provide a scatter-plot that shows the relation between computational cost (x-axis) and accuracy error (y-axis, log scale). Each dot is colored according to the pooling method, and we additionally vary the number K of filters, with K going from 2 to 20. Since the accuracy is monotone with K, one can additionally read the figure in K by going from top to bottom.

This scatter-plot provides additional evidence of our claim that the selective G-Bispectrum is faster than the G-TC while being more accurate than the Max G-Pooling and the Avg G-pooling. In terms of speed, we see that the selective G-Bispectrum provides an average gain of around 15 seconds over the G-TC, which corresponds to around 10% improvement. In terms of accuracy, we see that the selective G-Bispectrum provides an average gain of 0.02 accuracy over the Max G-pooling, and 0.26 over the Avg G-pooling.

We will include this plot in the main paper, and add corresponding plots for O2 and SO2/O2-EMNIST in the supplementary materials. We believe that they do, indeed, better quantify our claims. We thank the reviewers for the suggestion.

## Conclusion on experiments

We believe that the additional plots and proposed additional experiments on robutness strengthen our manuscript, and we thank the reviewers for their important feedback. Yet, we would like to emphasize that our work also makes several _theoretical_ contributions, and thus should not be reduced to its experiments. We believe that the impact of our reduction of the group-bispectrum is profound, with consequences for applied mathematics, signal processing and machine learning that go beyond classification results on benchmarks.

---

> ### Author Response · Authors · 2024-08-12
>
> Dear reviewers, please find below the results of an additionnal experiment.
>
> ### New Results on CIFAR-10
>
> Our results on CIFAR-10 present the same patterns as those on MNIST/EMNIST in the original manuscript. Here, we compare the performances of the G-Bispectrum layer with respect to the G-TC, the Max G-pooling and the Avg G-pooling models, trained on the SO(2)-CIFAR-10 datasets and we assess the accuracy by averaging the validation accuracy over 10 runs (see Table).
>
> The architecture of the neural network is kept minimal and is the same across experiments. As for our other experiments, we choose a simple architecture to isolate the contribution of the pooling layers. Here, we use a neural network with 1 convolution, 1 G-convolution and 1 G-pooling where the G-pooling varies across Avg G-Pooling, G-Bsp Max G-pooling and G-TC. We use $C_8$ and $D_8$ for the discretization of the groups. We use 16 filters.
>
> As for MNIST/EMNIST, the following patterns hold: at equivalent number of parameters, the more computationally expensive the pooling layer, the better the accuracy. The use of the G-TC becomes prohibitive when |G| increases.
>
> | Pooling | Avg acc | Std. dev. | K filters | Param. count|
> |---------|--------|------|--------|-------|
> | Avg | 56.49 | 0.77 | 16 | 1132255 |
> | BSP     | 71.51 | 3.38 | 16 | 1104025 |
> | Max     | 56.09 | 0.37 | 16 | 1132255 |
> | TC      | 80.99 | 4.20 | 16 | 1099040 |
>
> We hope that this additional result are convincing and motivate you to increase your score.

---

### Decision · Program_Chairs · 2024-09-25

**Decision:**

Accept (poster)

**Comment:**

The paper presents a new invariant layer for discrete group-convolutional networks based on the notion of the selective bispectrum. It tracks an earlier line of work in signal processing involving the notable works of Kakarala and Kondor, followed by the use of these ideas in the context of designing invariant layers in deep learning by Sanborn and Miolane. The advantage of using the triple correlation, and the bispectrum is that it is complete. But an issue with the use of the bispectrum is that it is expensive, involving $O(|G|^2)$ coefficients. The proposed selective bispectrum consists of only a subset i.e. $O(|G|)$ coefficients while still ensuring completeness. Since FFTs for groups of interest are available (or can be designed), this translates to a computation cost of $O(|G| log |G|)$. The paper supports the use of the selective bispectrum for designing invariant layers by some basic experiments which show that the proposed layer is faster than the bispectrum and TC layers (while trading off some robustness), and generally more accurate than a max-pooling layer.

As echoed by the reviewers, the paper is well-written and clearly motivated. More importantly, it does a good job at being self-contained, covering the necessary background on the triple correlation and the bispectrum. It also does a good job at covering existing work. The theoretical results basically involve a simple use of the Fourier transform and as such are straightforward, but are nevertheless novel and could also possibly be of interest to researchers in signal processing. Some of the reviewers were underwhelmed by the experiments. This stems from a valid concern since the method is aiming to be a replacement layer in deep neural networks. It is possible that the benefit from the proposed layer is most prominent in low data regimes. The authors have responded to the reviewers with some additional experiments on CIFAR-10, and providing computational tradeoffs. On considering these factors, I think the paper could still be a valuable contribution at NeurIPS. The paper's potential impact could be improved if it could also include results on a medical dataset (such as those used by Cohen and Winkels). The responses made to the reviewers should be incorporated in the paper for camera ready, inclduing more analysis such as the computation tradeoff. It is important to include computation metrics since it is presented as a selling point. It would also be useful to add a discussion about the use of the selective bispectrum in continuous groups. Note that in some networks on continuous groups, such as the Clebsch-Gordan Networks approach of Kondor, Lin, and Trivedi, an invariant layer is obtained by a projection. No explicit pooling is done. I wonder if the selective bispectrum still has an advantage in such situations. Finally, it would also be useful to spend some space to define the term "selective bispectrum" rather than leaving it implicit.